# Swap-guided Preference Learning for Personalized Reinforcement Learning from Human Feedback

**Gihoon Kim**[1]    **Euntai Kim**[1,2]

[1] Yonsei University    [2] Korea Institute of Science and Technology
{gihoon, etkim}@yonsei.ac.kr

## Abstract

Reinforcement Learning from Human Feedback (RLHF) is a widely used approach to align large-scale AI systems with human values. However, RLHF typically assumes a single, universal reward, which overlooks diverse preferences and limits personalization. Variational Preference Learning (VPL) seeks to address this by introducing user-specific latent variables. Despite its promise, we found that VPL suffers from posterior collapse. While this phenomenon is well known in VAEs, it has not previously been identified in preference learning frameworks. Under sparse preference data and with overly expressive decoders, VPL may cause latent variables to be ignored, reverting to a single-reward model. To overcome this limitation, we propose Swap-guided Preference Learning (SPL). The key idea is to construct fictitious swap annotators and use the mirroring property of their preferences to guide the encoder. SPL introduces three components: (1) swap-guided base regularization, (2) Preferential Inverse Autoregressive Flow (P-IAF), and (3) adaptive latent conditioning. Experiments show that SPL mitigates collapse, enriches user-specific latents, and improves preference prediction. Our code and data are available at https://github.com/cobang0111/SPL

## 1 Introduction

Reinforcement learning from human feedback (RLHF) has emerged as a prominent method for aligning large-scale AI systems with human values in various fields, particularly natural language processing (Ouyang et al., 2022). In RLHF, a reward model is first trained on human comparison data, and then a policy is optimized with reinforcement learning. This approach aligns model behavior more closely with human evaluations, improving performance, accuracy, and fairness across diverse domains (Leike et al., 2018; Ji et al., 2023).

However, most existing RLHF approaches (Christiano et al., 2017; Ouyang et al., 2022) are based on the single-reward assumption that all human preferences can be represented by a universal reward function. This assumption is originated from the Bradley–Terry–Luce (BTL) model (Bradley & Terry, 1952), which is commonly used to model pairwise comparisons and treats preferences as if they were generated from a shared scoring function. While mathematically convenient, this single-reward assumption is problematic in practice. Human preferences are not homogeneous but plural and often diverge across individuals or groups. Recent studies have shown that collapsing diverse perspectives into a single reward function introduces systematic bias in favor of majority preferences, overlooking groups and reducing fairness (Prabhakaran et al., 2021; Feffer et al., 2023; Casper et al., 2023). Consequently, models trained under this assumption may disadvantage underrepresented populations, even when their preferences are valid and important.

To address this issue, researchers have begun exploring what we refer to as personalized alignment (pluralistic alignment) (Sorensen et al., 2024). Instead of forcing all preferences into a single universal reward function, personalized alignment seeks to align different reward functions with different individuals according to their preferences, thereby capturing the heterogeneity of human values. One leading approach is Variational Preference Learning (VPL) (Poddar et al., 2024), which encodes

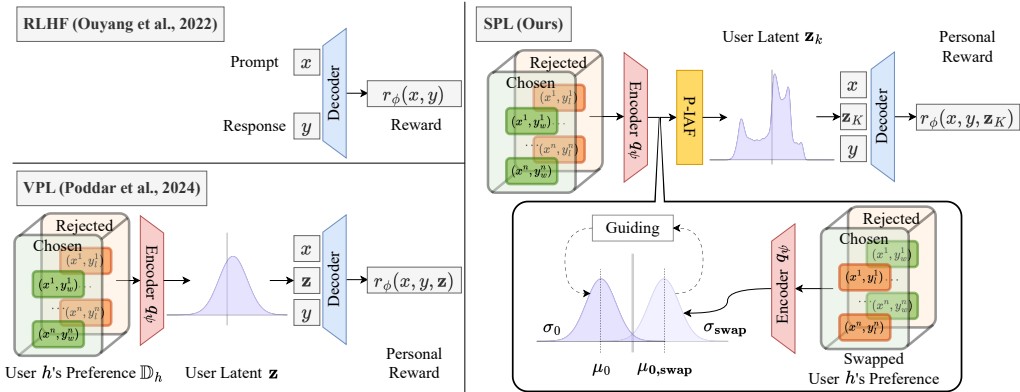

Figure 1: **Overview of SPL.** We propose *Swap-guided Preference Learning* (SPL), a new framework for personalized alignment. RLHF (Ouyang et al., 2022) cannot adequately reflect user diversity. To overcome this limitation, VPL (Poddar et al., 2024) encodes text data consisting of a prompt $x$ and response $y$ into a single latent embedding. However, this encoding process is highly prone to collapse. In contrast, SPL leverages the structural properties of preference data through guiding mechanisms and a Preferential Inverse Autoregressive flow, allowing the latent space to capture user-specific characteristics.

user-specific latent variables from preference data and decodes them into corresponding rewards. This framework allows AI systems to flexibly adapt to diverse users without relying on predefined groupings or rigid categorization.

Despite its promise, we found in our experiments that VPL suffers from practical failure mode: posterior collapse. This phenomenon is sometimes observed in VAEs (Bowman et al., 2016; Chen et al., 2016; He et al., 2019; Lucas et al., 2019; Wang et al., 2021) but has not previously been identified in preference learning frameworks. When combined with a strong reward decoder, this posterior collapse can cause the encoder's latent variable to become uninformative and effectively ignored. In such cases, the latent variable fails to capture user-specific information, and the decoder explains preferences without relying on it. Training then reduces to an implicit single reward model, ignoring minority preferences and undermining the goal of personalized alignment.

To overcome this, we introduce *Swap-guided Preference Learning* (SPL), an expressive variational framework for personalized alignment that explicitly leverages the structural properties of preference pair data. To the best our knowledge, we are the first to report and address posterior collapse in preference learning. Our approach improves user-latent encoding and reward decoding through three key innovations: (i) **Swap-guided Base Regularization**, which encouraging latent space shows *mirrored* characteristics under preference swapping; (ii) **Preferential-Inverse Autoregressive Flow**, which disentangles swap-reversal and swap-invariant signals, conditioning a inverse autoregressive flow on them to yield improved latent representations without collapse; and (iii) **Adaptive Latent Conditioning**, which dynamically adjusts the contribution of the latent variable to reward prediction. Together, these mechanisms consistently reduce posterior collapse and enable more faithful and pluralistic preference modeling.

## 2    PRELIMINARY FUNDAMENTALS

**Reinforcement Learning from Human Feedback**    For post-training of Large Language Models (LLM), RLHF relies on a dataset of $N$ human preference pairs, $\mathbb{D} = \{(x^i, y_w^i, y_l^i)\}_{i=1}^N$, where $x$ is a prompt and $(y_w, y_l)$ denote the chosen(winning) and rejected(losing) responses, respectively. RLHF assumes an single universal reward function $r_\phi(x, y)$, optimized by maximizing the log-likelihood of observed preferences:

$$\mathbb{E}_{(x,y_w,y_l)\sim\mathbb{D}}\Big[\log p_\phi(y_w \succ y_l \mid x)\Big]. \tag{1}$$

The preference probability $p_\phi(y_w \succ y_l \mid x)$ is typically modeled via the Bradley–Terry–Luce (BTL) model (Bradley & Terry, 1952):

$$p_\phi(y_w \succ y_l \mid x) = \frac{\exp\big(r_\phi(x, y_w)\big)}{\exp\big(r_\phi(x, y_w)\big) + \exp\big(r_\phi(x, y_l)\big)} = \sigma\big(r_\phi(x, y_w) - r_\phi(x, y_l)\big), \qquad (2)$$

where $\sigma$ denotes the logistic function. Thus, the reward function $r_\phi$ is trained to explain human-preferred outcomes, and the learned reward model is subsequently used to optimize a policy aligned with human judgments.

**Variational Approach for Personalized Alignment**  A central direction in personalized alignment is to condition reward models and policies on user-specific information (Oh et al., 2024; Poddar et al., 2024; Bose et al., 2025; Shenfeld et al., 2025; Gong et al., 2025). Among these approaches, Variational Preference Learning (VPL) (Poddar et al., 2024) is particularly influential. Inspired by variational autoencoders (VAEs) (Kingma et al., 2013), VPL introduces a user-specific latent variable $z \in \mathbb{R}^d$ inferred from each user $h$'s preference dataset $\mathbb{D}_h = \{(x^i, y_w^i, y_l^i)\}_{i=1}^n \subset \mathbb{D}$. The encoder produces an approximate posterior $q_\psi(z \mid \mathbb{D}_h)$, while the decoder predicts rewards for prompt–response pairs $(x, y)$ conditioned on $z$, denoted as $r_\phi(x, y, z)$.

Formally, VPL extends the objective in Eq.(1) by adding a variational regularization term. This yields an evidence lower bound (ELBO):

$$\mathbb{E}_{h \sim \mathbb{H}}\left[\mathbb{E}_{\substack{z \sim q_\psi(z \mid \mathbb{D}_h) \\ (x, y_w, y_l) \sim \mathbb{D}_h}} \big[\log p_\phi(y_w \succ y_l \mid x, z)\big] - \beta D_{\mathrm{KL}}\big[q_\psi(z \mid \mathbb{D}_h)\|p(z)\big]\right], \qquad (3)$$

where $\beta$ is KL divergence weight and the $\log p(z)$ represents the prior distribution's log-density, selected as $\mathcal{N}(\mathbf{0}, \mathbf{I})$. This objective maximizes the conditional log-likelihood of preferences while regularizing the user-specific posterior toward the prior, thereby preventing overfitting and encouraging generalizable latent structure. By leveraging $z$, VPL provides flexibility in modeling personalized traits and has shown strong empirical performance in capturing diverse preferences. However, recent work (Nam et al., 2025) indicates that compressing rich textual preference data into a single latent embedding $z$ remains highly challenging.

**Inverse Autoregressive Flow**  Normalizing flows (Rezende & Mohamed, 2015) is a framework for constructing flexible posterior distributions by applying a sequence of invertible transformations. Among them, Inverse Autoregressive Flow (IAF) (Kingma et al., 2016) is specifically designed to enrich the expressivity of variational posteriors while preserving computational tractability. The procedure begins with a base latent variable $z_0 \in \mathbb{R}^d$ and context vector $c \in \mathbb{R}^{d_c}$ drawn from encoder (i.e., $q_\psi(z_0 \mid x) = \mathcal{N}(\mu, \sigma^2)$ with additional output $c$), followed by a series of parameterized, invertible transformations $f_k$. After $K$ step transformations, the final variable $z_K$ acquires a more complex distribution:

$$z_0 \sim q_\psi(z_0 \mid x), \quad z_k = f_k(z_{k-1}, c), \quad k = 1, \ldots, K$$

When each $f_k$ admits a tractable Jacobian determinant, the density of $z_K$ can be computed efficiently via the change-of-variables formula:

$$\log q_\psi(z_K \mid x) = \log q_\psi(z_0 \mid x) - \sum_{k=1}^{K} \log \det \left|\frac{\partial z_k}{\partial z_{k-1}}\right|. \qquad (4)$$

In practice, IAF employs autoregressive neural networks to parameterize shift and scale functions:

$$z_k = \mu_k(z_{k-1}, c) + \sigma_k(z_{k-1}, c) \odot z_{k-1}, \qquad (5)$$

where $\mu_k$ and $\sigma_k$ are autoregressively conditioned on the preceding dimensions of $z_{k-1}$. This autoregressive structure ensures a lower-triangular Jacobian, making the determinant easy to compute:

$$\log \det \left|\frac{\partial z_k}{\partial z_{k-1}}\right| = \sum_{j=1}^{d} \log \left|\sigma_k^j\right|, \qquad (6)$$

with $\sigma_k^j$ denoting the $j$-th element of the scale function.

As a result, IAF enables parallelizable sampling and yields a substantially richer posterior $q_\psi(z_K \mid x)$ that captures inter-dimensional dependencies and non-Gaussian structures (e.g., skewness, heavy tails) beyond the capacity of the base posterior (Kingma et al., 2016; Papamakarios et al., 2021).

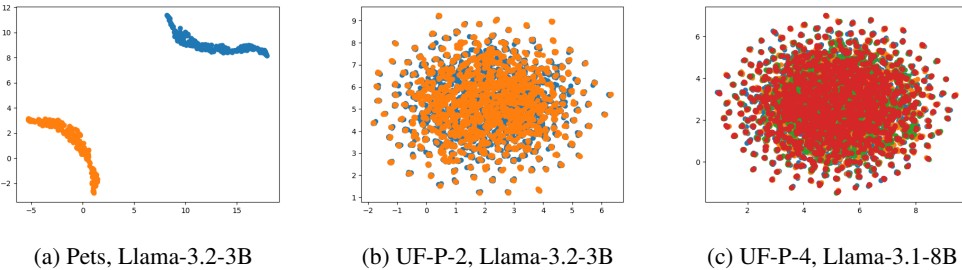

| (a) Pets, Llama-3.2-3B | (b) UF-P-2, Llama-3.2-3B | (c) UF-P-4, Llama-3.1-8B |

Figure 2: **Posterior collapse in Variational Preference Learning.** We visualize latent embeddings $z$ from the VPL encoder using 2D UMAP (McInnes et al., 2018). Each point denotes a user, colored by their preference type. (a) User preference types are distinctly separated, indicating non-collapse. (b), (c) Latent collapse occurs, making preference types indistinguishable.

## 3 MOTIVATION

In this section, we explain the posterior collapse that we observed in preference learning and identify some guidance by comparing collapse and non-collapse cases. Fig. 2 illustrates this phenomenon that we observed in VPL. The simple dataset *Pets* simulates multi-modal user preferences over pets (e.g., dog, cat) given a single shared prompt. In contrast, the more complex *UF-P* datasets contain 2 or 4 preference types (e.g., helpfulness, honesty, instruction-following, and truthfulness) and include diverse prompts and responses. In Fig. 2a, two user types (in different colors) are clearly separated in the latent space for the *Pets*. However, in *UF-P*, users merge into a single cluster, losing separation as shown in Fig. 2b and 2c.

This collapse appears to stem from two factors: (1) noisy and ambiguous human feedback, together with the difficulty of compressing diverse, complex textual preferences in the encoder, often leads to unstable latent learning, which in turn causes the reward decoder to ignore the $z$ pathway; and (2) the reward decoder already receives sufficient information from the complete prompt–response pair, allowing it to maximize the likelihood in Eq.(3) without relying on $z$. This leads to the latent variable failing to capture user-specific information and becoming uninformative. Further evidence of posterior collapse is presented in Appendix A.

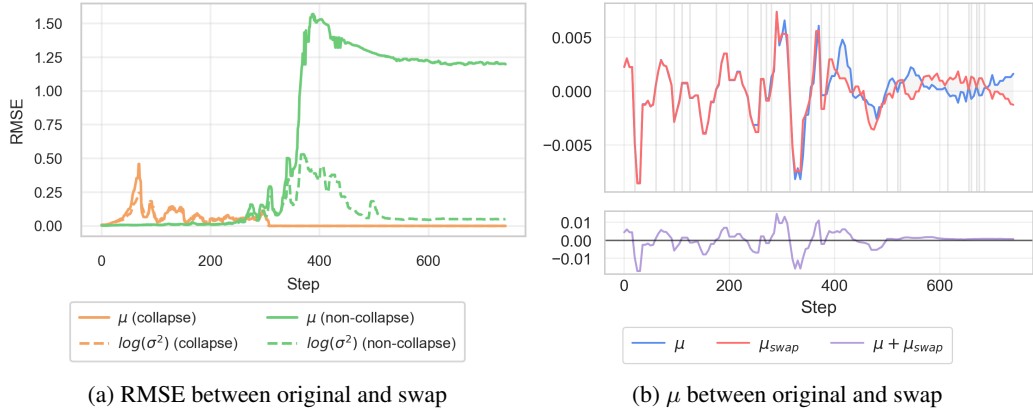

| (a) RMSE between original and swap | (b) $\mu$ between original and swap |

Figure 3: **Differences in posterior distribution between original and swapped inputs.** We test how the VPL encoder's posterior responds when each preference pair is inverted to simulate a user with opposite choices, using the simple dataset *Pets*. (a) Average RMSE between original and swapped inputs across posterior mean $\mu$ and log-variance $\ell$. Collapse appears in Llama-3.1-8B (orange), where both parameters remain unchanged, whereas Llama-3.2-3B (green) shows distinct behavior. (b) Plot $\mu$ vs. $\mu_{\text{swap}}$ for Llama-3.2-3B; $\mu+\mu_{\text{swap}}$ is in the lower panel. Initially, the curves are similar, but their difference grows and stabilizes as learning continues, resulting in a sign-reversal.

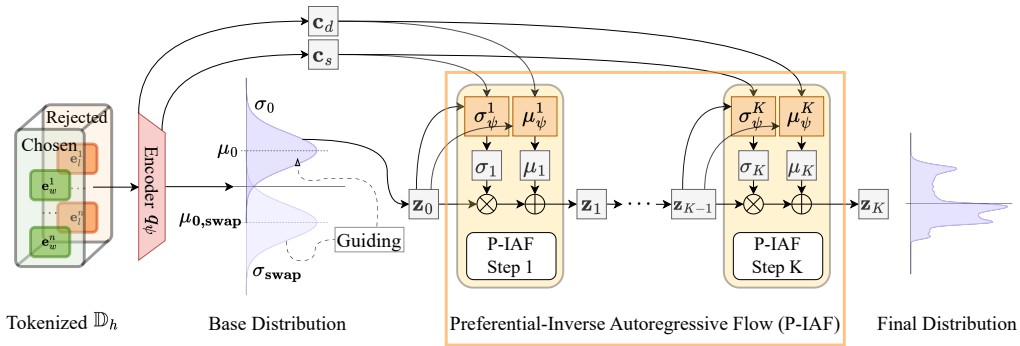

Figure 4: Preference encoding process of SPL

To address the posterior collapse in VPL, we examine the information captured in the posterior distribution when user preferences are successfully encoded, and use this insight to guide the design of an effective user-latent space. To this end, we conduct a simple swap experiment. For a user $h$ with dataset $\mathbb{D}_h$, suppose the encoder outputs $q_\psi(\boldsymbol{z} \mid \mathbb{D}_h) = \mathcal{N}(\boldsymbol{\mu}, \boldsymbol{\sigma}^2)$, where $\boldsymbol{\mu}, \boldsymbol{\sigma}^2 \in \mathbb{R}^d$. We then construct a fictitious user $h_{\text{swap}}$ with the opposite preference of $h$ by swapping the chosen and rejected responses in every pair, as shown in the right part of Fig. 1. Feeding these swapped pairs into the encoder yields $q_\psi(\boldsymbol{z} \mid \mathbb{D}_{h_{\text{swap}}}) = \mathcal{N}(\boldsymbol{\mu}_{\text{swap}}, \boldsymbol{\sigma}^2_{\text{swap}})$. Fig. 3a visualizes the RMSE between $\boldsymbol{\mu}$ and $\boldsymbol{\mu}_{\text{swap}}$, and between $\boldsymbol{\ell} = \log \boldsymbol{\sigma}^2$ and $\boldsymbol{\ell}_{\text{swap}} = \log \boldsymbol{\sigma}^2_{\text{swap}}$, over the course of training for both collapse and non-collapse cases. In the collapse case, the RMSE converges to zero for both $\boldsymbol{\mu}$ and $\boldsymbol{\ell}$, i.e., $\boldsymbol{\mu} \approx \boldsymbol{\mu}_{\text{swap}}$ and $\boldsymbol{\ell} \approx \boldsymbol{\ell}_{\text{swap}}$, indicating that the latent variable carries no user-specific signal and is effectively ignored by the decoder. In the non-collapse case, however, the RMSE of $\mu$ converges to a non-zero value, implying a clear separation between the original user $h$ and the fictitious user $h_{\text{swap}}$. In particular, $\boldsymbol{\mu}$ and $\boldsymbol{\mu}_{\text{swap}}$ exhibit a sign-reversal, $\boldsymbol{\mu} \approx -\boldsymbol{\mu}_{\text{swap}}$, as shown in Fig. 3b, while the log-variance remains invariant to swaps, $\boldsymbol{\ell} \approx \boldsymbol{\ell}_{\text{swap}}$, i.e., the posterior distribution exhibits a "*mirrored*" distribution when swapped. This structural division implies that $\boldsymbol{\mu}$ captures swap-reversal information, whereas $\boldsymbol{\ell}$ captures swap-invariant information. Such disentanglement makes the latent variable essential for the decoder. In the next section, we use this insight to develop our new preference learning framework.

# 4 METHOD

We propose *Swap-guided Preference Learning* (SPL), a new framework for preference learning that regularizes the encoder with guidance from preference swapping. This approach consistently reduces posterior collapse while ensuring that user-specific information is faithfully encoded in the latent variable $\boldsymbol{z}$. To achieve this, we introduce three components: (i) Swap-guided Base Regularization, (ii) Preferential Inverse Autoregressive Flow (P-IAF), and (iii) Adaptive Latent Conditioning.

## 4.1 ENCODING USER PREFERENCES INTO A LATENT

To encourage our SPL to encode user preferences into a latent $\boldsymbol{z}$, we introduce two strategies in this section. The first is to enforce the output from the encoder to satisfy the mirroring of preference swaps, thereby mitigating posterior collapse. We call the encoder's Gaussian output is termed the base distribution and denoted as $\boldsymbol{z}_0$. The second strategy is to transform $\boldsymbol{z}_0$ using an Inverse Autoregressive Flow (IAF), warping the Gaussian $\boldsymbol{z}_0$ into a richer distribution $\boldsymbol{z}_K$. In this second strategy, we also control the flow from the base $\boldsymbol{z}_0$ to the transformed distribution $z_K$ with guidance from the mirroring of preference swaps. The two strategies are illustrated in Fig. 4. We now explain them one by one.

**Swap-guided Base Regularization** Based on the mirroring of preference swaps in section 3, the encoder is trained to learn user preferences by generating mirrored distributions for annotators $h$ and $h_{\text{swap}}$. Specifically, given an annotator $h$ and its fictitious opposite annotator $h_{\text{swap}}$, with encoder

outputs $\mathcal{N}(\boldsymbol{\mu}, \boldsymbol{\ell})$ and $\mathcal{N}(\boldsymbol{\mu}_{\text{swap}}, \boldsymbol{\ell}_{\text{swap}})$, respectively, we train the encoder so that the two means $\boldsymbol{\mu}$ and $\boldsymbol{\mu}_{\text{swap}}$ exhibit a sign-reversal, while the two log-variances $\boldsymbol{\ell}$ and $\boldsymbol{\ell}_{\text{swap}}$ remain invariant. This is achieved by applying the guidance loss $\mathcal{L}_{\text{guide}}$ defined by

$$\cos(\boldsymbol{\mu}, \boldsymbol{\mu}_{\text{swap}}) = \frac{\boldsymbol{\mu}^\top \boldsymbol{\mu}_{\text{swap}}}{(\|\boldsymbol{\mu}\| + \varepsilon)(\|\boldsymbol{\mu}_{\text{swap}}\| + \varepsilon)}, \quad \cos(\boldsymbol{\ell}, \boldsymbol{\ell}_{\text{swap}}) = \frac{\boldsymbol{\ell}^\top \boldsymbol{\ell}_{\text{swap}}}{(\|\boldsymbol{\ell}\| + \varepsilon)(\|\boldsymbol{\ell}_{\text{swap}}\| + \varepsilon)},$$

and define the encoder $q_\psi$ training guidance loss as:

$$\mathcal{L}_{\text{guide}} = \mathbb{E}_{h \sim \mathbb{H}}\left[ \tfrac{1}{2}\big(1 + \cos(\boldsymbol{\mu}^h, \boldsymbol{\mu}_{\text{swap}}^h)\big) + \eta\, \tfrac{1}{2}\big(1 - \cos(\boldsymbol{\ell}^h, \boldsymbol{\ell}_{\text{swap}}^h)\big) \right]. \tag{7}$$

$\eta$ balances mean and variance; $\varepsilon > 0$ ensures stability.

**Preferential Inverse Autoregressive Flow**  The next step is to apply IAF to warp the Gaussian $\boldsymbol{z}_0$ into a multi-modal distribution $\boldsymbol{z}_K$. Unlike the base regularization, we cannot enforce the mirroring property of preference swaps in this transformation, because $\boldsymbol{z}_K$ is no longer Gaussian and cannot be characterized in terms of mean and variance. In other words, the flow from $\boldsymbol{z}_0$ to $\boldsymbol{z}_K$ under a standard IAF cannot be directly controlled to satisfy the mirroring property of preference swaps. To address this limitation, we propose Preferential Inverse Autoregressive Flow (P-IAF), which decomposes the context vector $\boldsymbol{c}$ into swap-reversal and swap-invariant components. Intuitively, the swap-reversal context $\boldsymbol{c}_d$ captures the directional preference signals that reflect the mirroring of swaps, while the swap-invariant context $\boldsymbol{c}_s$ captures the background information. Our P-IAF is defined by

$$\boldsymbol{z}_k = f_\psi^k(\boldsymbol{z}_{k-1}, \boldsymbol{c}_d, \boldsymbol{c}_s) = \mu_k(\boldsymbol{z}_{k-1}, \boldsymbol{c}_d) + \sigma_k(\boldsymbol{z}_{k-1}, \boldsymbol{c}_s) \odot \boldsymbol{z}_{k-1}, \tag{8}$$

where $k = 1, \ldots, K$. We form $\boldsymbol{c}_d$ and $\boldsymbol{c}_s$ by a swap-reversal and swap-invariant decomposition of the encoder's additional output $\boldsymbol{c}$ (from $\mathbb{D}_h$) and $\boldsymbol{c}_{\text{swap}}$ (from swapped counterpart $\mathbb{D}_{h_{\text{swap}}}$) as follows:

$$\boldsymbol{c}_d \triangleq \tfrac{1}{2}(\boldsymbol{c} - \boldsymbol{c}_{\text{swap}}), \quad \boldsymbol{c}_s \triangleq \tfrac{1}{2}(\boldsymbol{c} + \boldsymbol{c}_{\text{swap}}),$$

which guarantees $\boldsymbol{c} = \boldsymbol{c}_d + \boldsymbol{c}_s$, $\boldsymbol{c}_{\text{swap}} = -\boldsymbol{c}_d + \boldsymbol{c}_s$. By feeding $\boldsymbol{c}_d$ only to the shift function $\mu_k$ and $\boldsymbol{c}_s$ only to the scale function $\sigma_k$, P-IAF reduces cross-context coupling between swap-reversal and swap-invariant signals, thereby preserving pair-derived user preference more effectively while retaining IAF's expressivity from the autoregressive composition. See Appendix B for details and proof.

Substituting Eq.(8) into Eq.(4) yields the overall log posterior after $K$ flow steps

$$\log q_\psi(\boldsymbol{z}_K \mid \mathbb{D}_h) = \log q_\psi(\boldsymbol{z}_0 \mid \mathbb{D}_h) - \sum_{k=1}^{K} \sum_{j=1}^{d} \log |\sigma_k^j|, \tag{9}$$

and the KL divergence of Eq.(3) is given by[1]:

$$D_{\text{KL}} = \mathbb{E}_{h \sim \mathbb{H}}\big[\log q_\psi(\boldsymbol{z}_K \mid \mathbb{D}_h) - \log p(\boldsymbol{z}_K)\big], \tag{10}$$

where $\log q_\psi(\boldsymbol{z}_K \mid \mathbb{D}_h)$ is given in Eq.(9).

## 4.2 Decoding personalized rewards from latents

The decoder scores a prompt–response $(x, y)$ conditioned on the user-latent $\boldsymbol{z}_K$, yielding $r_\phi(x, y, \boldsymbol{z}_K)$, and is trained to satisfy $r_\phi(x, y_w, \boldsymbol{z}_K) > r_\phi(x, y_l, \boldsymbol{z}_K)$. Extending Eq.(2) about $\boldsymbol{z}_K$, the decoder training objective over users $h \sim \mathbb{H}$:

$$\mathbb{E}_{h \sim \mathbb{H}}\left[ \mathbb{E}_{\substack{\boldsymbol{z}_K \sim q_\psi(\boldsymbol{z}_K \mid \mathbb{D}_h) \\ (x, y_w, y_l) \sim \mathbb{D}_h}} \big[\log p_\phi(y_w \succ y_l \mid x, \boldsymbol{z}_K)\big] \right] \tag{11}$$

where $p_\phi(y_w \succ y_l \mid x, \boldsymbol{z}_K) = \sigma\big(r_\phi(x, y_w, \boldsymbol{z}_K) - r_\phi(x, y_l, \boldsymbol{z}_K)\big)$ which means preference probability conditioned on $\boldsymbol{z}_K$.

---

[1]For notational simplicity, we denote all learnable parameters by $\psi$. In practice, $\psi$ includes both (i) encoder parameters $\psi_{\text{enc}}$ and (ii) flow parameters $\psi_{\text{flow}} = \{\psi_{\mu_k}, \psi_{\sigma_k}\}_{k=1}^{K}$ corresponding to the shift and scale function in each flow transformation step $k$.

**Adaptive Latent Conditioning**   Inspired by feature modulation (Perez et al., 2018), we design a per-user modulation decoder that adapts prompt-response embeddings based on the user-latent embedding $z_K$, allowing dynamic influence adjustment when predicting input rewards. For example, when the latent embedding provides strong signals of user preference, its contribution to reward prediction is amplified, whereas when the preference signal is uncertain, the contribution is attenuated. Detailed modeling of this adaptive conditioning mechanism is provided in Appendix C.

### 4.3 OBJECTIVE FUNCTION OF SPL

Maximize the log-likelihood term from Eq.(11) while minimizing the KL divergence term in Eq.(10), the ELBO of SPL is defined across the entire user $\mathbb{H}$ as:

$$\text{ELBO} = \mathop{\mathbb{E}}_{h \sim \mathbb{H}} \left[ \mathop{\mathbb{E}}_{\substack{z_K \sim q_\psi(z_K | \mathbb{D}_h) \\ (x, y_w, y_l) \sim \mathbb{D}_h}} [\log p_\phi(y_w \succ y_l \mid x, z_K)] - \beta(\log q_\psi(z_K \mid \mathbb{D}_h) - \log p(z_K)) \right] \quad (12)$$

We regularize the base posterior $q_\psi(z_0 \mid \mathbb{D}_h)$ using the guidance loss in Eq.(7). The final objective minimizes:

$$\mathcal{L}(\phi, \psi) = -\text{ELBO} + \lambda \mathcal{L}_{\text{guide}} \quad (13)$$

where $\lambda$ controls the strength of the guidance loss term. Consequently, the reward model explicitly conditions on the user-latent $z_K$, yielding a personalized reward $r_\phi(\cdot, \cdot, z_K)$; optimizing the policy under this reward personalizes behavior and thus achieves personalized alignment.

## 5 EXPERIMENTS

In this section, we evaluate the performance of SPL. First, we examine whether SPL can construct a meaningful latent space without posterior collapse. Second, we evaluate whether SPL effectively improves preference-prediction accuracy. SPL remains stable across different KL divergence weights $\beta$, unlike the earlier approach (Poddar et al., 2024) in our experiments. Moreover, SPL consistently outperforms baselines in preference-prediction accuracy. Before presenting these results, we describe our experimental setup.

**Baselines**   We compare our method against the following baselines:

- **BTL** (Ouyang et al., 2022): The standard RLHF based on Bradley–Terry–Luce model.
- **DPL** (Siththaranjan et al., 2023): Distributional Preference Learning, which captures implicit context across the entire preference dataset and models the reward as a distribution but doesn't consider individual user preferences.
- **VPL** (Poddar et al., 2024): Variational Preference Learning, which employs the user-latent embedding with a simple Gaussian posterior distribution, without swap-guided encoding and latent conditioning.
- **SPL** (Ours): Our proposed method.

For all methods, we use supervised fine-tuned LLMs based on *Llama-3* (Dubey et al., 2024), specifically two variants: *Llama-3.2-3B* and *Llama-3.1-8B*.

**Datasets**   We conduct experiments on two datasets: a simple preference dataset *Pets* and a complex preference dataset *UltraFeedback-P (UF-P)* (Poddar et al., 2024) derived from *Ultrafeedback* (Cui et al., 2023), featuring user types pursuing values like helpfulness, honesty, instruction-following, and truthfulness.

The *Pets* dataset uses a single shared prompt, "Please talk about one kind of pets." and response is a description of one of four animals (dog, cat, rabbit and bird). The dataset defines two user types that agree on the most- and least-preferred animals (bird and rabbit, respectively) but disagree on the relative ordering of the middle options (dog vs. cat), inducing a multi-modal distribution over user preferences.

The *UF-P* dataset assumes that each user $h$ belongs to one of several preference types $\mathbb{P}$ (e.g., $p \in \mathbb{P} = \{\text{helpfulness, honesty}\}$). It is constructed from the *Ultrafeedback* prompt–response data, where responses are labeled by *GPT-4* (Achiam et al., 2023) with scores for each type $p$. For each prompt, the winning and losing responses are selected according to the score associated with the target preference type $p$. Specifically, *UF-P-2* contains two preference types focusing on helpfulness and honesty, while *UF-P-4* contains four preference types focusing on helpfulness, honesty, instruction-following, and truthfulness. Due to its diverse preference modes and a wide variety of prompt–response pairs, the *UF-P* dataset is highly ambiguous and challenging.

In all datasets, one sample $\mathbb{D}_h$ corresponds to a user $h$ with a user type $p$. This type information is used only when constructing $\mathbb{D}_h$ (to determine winning and losing responses from *Ultrafeedback*) and for qualitative evaluation (to verify whether user types are well-separated). Importantly, the latent embedding relies on user preference data $\mathbb{D}_h$, not on type $p$. Additional details about experiments are provided in Appendix E.

## 5.1 RESULTS

We first demonstrate that our method effectively reduces posterior collapse and encodes a stable latent space. To diagnose collapse quantitatively, we evaluate using the *Active Units* (AU) metric from prior work (Burda et al., 2015). AU counts latent dimensions with variability exceeds a small threshold $\delta$; a dimension $u$ is considered active if its posterior mean responses show sufficient variability across the evaluation set $\mathbb{D}_{\text{eval}}$.

$$AU = |\{u : \text{Var}_{\mathbb{D}_{\text{eval}}}\big(\mu_{\psi,u}(\mathbb{D}_{\text{eval}})\big) > \delta\}|$$

Thus, $AU = 0$ means all latent dimensions are unresponsive across evaluation data. In these runs, the encoder outputs fixed posterior means and variances with $AU = 0$, indicated as *posterior collapse* and shaded gray in tables. Accuracy is the ratio of evaluation samples where predicted rewards match user preferences (i.e., winning responses have higher rewards).

Table 1: Accuracy and active units across $\beta$

| Model | $\beta$ | Method | UF-P-2 | | UF-P-4 | |
|---|---|---|---|---|---|---|
| | | | Acc. [%] | AU [%] | Acc. [%] | AU [%] |
| Llama-3.2-3B | $3 \times 10^{-7}$ | VPL | $62.04 \pm 0.16$ | $0.00 \pm 0.00$ | $56.91 \pm 0.12$ | $0.00 \pm 0.00$ |
| | | SPL | $62.59 \pm 0.37$ | $73.05 \pm 4.69$ | $61.52 \pm 0.27$ | $76.89 \pm 7.89$ |
| | $3 \times 10^{-6}$ | VPL | $62.37 \pm 0.15$ | $88.22 \pm 7.72$ | $57.03 \pm 0.10$ | $0.00 \pm 0.00$ |
| | | SPL | $\mathbf{63.28 \pm 0.13}$ | $\mathbf{93.07 \pm 3.18}$ | $61.56 \pm 0.03$ | $\mathbf{82.32 \pm 3.18}$ |
| | $3 \times 10^{-5}$ | VPL | $62.29 \pm 0.19$ | $14.03 \pm 6.93$ | $57.00 \pm 0.07$ | $0.00 \pm 0.00$ |
| | | SPL | $62.69 \pm 0.20$ | $70.05 \pm 8.15$ | $\mathbf{61.62 \pm 0.18}$ | $77.15 \pm 9.12$ |
| Llama-3.1-8B | $3 \times 10^{-7}$ | VPL | $62.46 \pm 0.08$ | $0.00 \pm 0.00$ | $57.25 \pm 0.22$ | $0.00 \pm 0.00$ |
| | | SPL | $63.58 \pm 0.14$ | $92.19 \pm 2.25$ | $61.92 \pm 0.08$ | $93.75 \pm 4.67$ |
| | $3 \times 10^{-6}$ | VPL | $62.66 \pm 0.23$ | $91.05 \pm 4.43$ | $57.14 \pm 0.05$ | $0.00 \pm 0.00$ |
| | | SPL | $\mathbf{63.71 \pm 0.18}$ | $\mathbf{97.10 \pm 1.14}$ | $62.21 \pm 0.06$ | $\mathbf{96.19 \pm 2.33}$ |
| | $3 \times 10^{-5}$ | VPL | $62.54 \pm 0.15$ | $90.79 \pm 7.96$ | $57.18 \pm 0.11$ | $0.00 \pm 0.00$ |
| | | SPL | $63.43 \pm 0.04$ | $90.07 \pm 2.98$ | $\mathbf{62.46 \pm 0.07}$ | $85.90 \pm 3.40$ |

Table 1 shows results for VPL and SPL across a range of KL-divergence weights $\beta$ under three distinct random seeds. Prior methods require careful tuning of $\beta$ to avoid collapse. In contrast, SPL exhibits no posterior collapse in any of the tested settings. The advantage is most evident on highly multi-modal preference datasets *UF-P-4*, where VPL collapses under all tested $\beta$ values, but SPL consistently maintains high AU. Notably, SPL is much less sensitive to $\beta$.

Table 2: Preference-prediction accuracy (%) compared with baselines

| Model | Method | Pets | UF-P-2 | UF-P-4 |
|---|---|---|---|---|
| Llama-3.2-3B | BTL | $57.48 \pm 2.37$ | $62.25 \pm 0.03$ | $57.07 \pm 0.01$ |
| | DPL | $62.02 \pm 1.92$ | $62.22 \pm 0.03$ | $57.04 \pm 0.05$ |
| | VPL | $99.67 \pm 0.38$ | $62.37 \pm 0.15$ | $57.03 \pm 0.10$ |
| | SPL (Ours) | $\mathbf{100.0 \pm 0.00}$ | $\mathbf{63.28 \pm 0.13}$ | $\mathbf{61.56 \pm 0.03}$ |
| Llama-3.1-8B | BTL | $60.74 \pm 0.49$ | $62.59 \pm 0.04$ | $57.40 \pm 0.28$ |
| | DPL | $61.03 \pm 0.25$ | $62.74 \pm 0.03$ | $57.66 \pm 0.14$ |
| | VPL | $75.33 \pm 0.63$ | $62.66 \pm 0.23$ | $57.14 \pm 0.05$ |
| | SPL (Ours) | $\mathbf{100.0 \pm 0.00}$ | $\mathbf{63.71 \pm 0.18}$ | $\mathbf{62.21 \pm 0.06}$ |

Next, Table 2 compares preference-prediction accuracy against baselines. For fairness, we fix $\beta = 3 \times 10^{-6}$—a setting under which VPL is comparatively more stable—and report the mean $\pm$ standard deviation over three distinct random seeds for all methods. Across all datasets and models, SPL achieves higher preference-prediction accuracy than competing baselines. These results mean swap-guided base regularization and P-IAF are effectively encoding user preference to identifiable user-latent. Furthermore, SPL improves accuracy and prevents collapse without requiring substantial additional computation or memory. We measured training computation and memory costs on the *UF-P-4* dataset. As shown in the Table 3, SPL achieves these gains with only minimal computational and memory overhead.

Table 3: Training computation and memory costs on UF-P-4

| Model | Method | sample/s | GPU hour | peak memory |
|---|---|---|---|---|
| Llama-3.2-3B | VPL | 6.070 | 13.363 | 6.25GB |
| | SPL | 5.952 | 13.590 | 6.65GB |
| Llama-3.1-8B | VPL | 3.370 | 23.791 | 11.37GB |
| | SPL | 3.355 | 23.945 | 11.76GB |

We further examine the learned latent spaces qualitatively. Fig. 5 visualizes the encoded user-latent for the *UF-P* dataset using Llama-3.1-8B. SPL yields more compact and distinctly separated embeddings compared to VPL. This shows that swap-guided base regularization and P-IAF of SPL effectively encode user preferences in the latent space. Further analysis of additional experiments is provided in Appendix D.

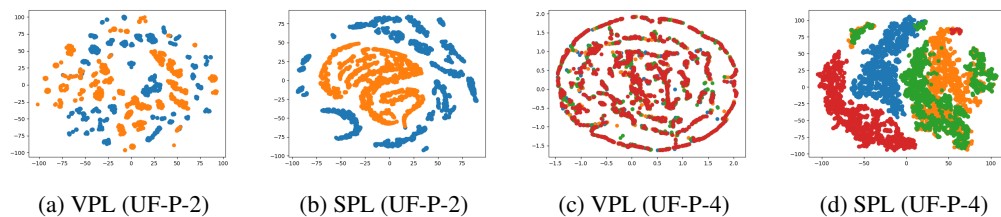

(a) VPL (UF-P-2)  (b) SPL (UF-P-2)  (c) VPL (UF-P-4)  (d) SPL (UF-P-4)

Figure 5: **Latent embeddings learned on the *UF-P* dataset.** We visualize latent embeddings $z$ from baselines and SPL (Ours) encoder using 2D t-SNE (Maaten & Hinton, 2008). Each point denotes a user, colored by their preference type. Compared to the VPL, SPL yields much clearer separation in the latent space.

## 6    CONCLUSION

We proposed *Swap-guided Preference Learning* (SPL), a framework that overcomes the failure mode in preference learning on complex textual preference data. Across all experiments, SPL consistently

improves prediction accuracy over baselines and prevents collapse. These results suggest that combining our swap-guided base regularization, P-IAF, and adaptive latent conditioning effectively encodes user-specific latents from complex textual preferences, even with sparse preference signals. Consequently, by explicitly conditioning the reward on the user latent and optimizing the policy under this reward, our framework enables user-specific behaviors, achieving personalized alignment.

**Limitation** Our study focuses on encoding user preferences from independent, single-turn comparison data. This data requirement can be burdensome and may feel unnecessary from a user perspective. We believe our framework can be extended to preferences expressed over natural, multi-turn dialogue; we consider this for future work.

ACKNOWLEDGMENTS

This research was supported by the KIST Institutional Program (2E33801).

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

APPENDIX

## A    EVIDENCE FOR COLLAPSE VIA POSTERIOR–PRIOR RESPONSE

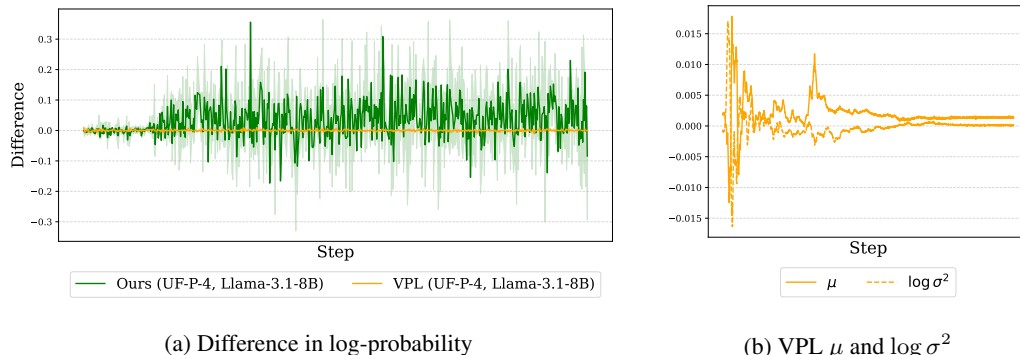

(a) Difference in log-probability

(b) VPL $\mu$ and $\log \sigma^2$

Figure 6: Evidence of posterior collapse in preference learning

We provide evidence that VPL causes the latent to collapse, preventing meaningful encoding during training. Fig. 6a contrasts decoder outputs from a approximate posterior latent and a noise vector $\epsilon$ from the prior $\mathcal{N}(\mathbf{0}, \mathbf{I})$. Specifically, we compute

$$\left[\log p_\phi(y_w \succ y_l \mid x, \mathbf{z}) - \log p_\phi(y_w \succ y_l \mid x, \boldsymbol{\epsilon})\right].$$

In non-collapsing runs (e.g., with our SPL), the difference persists, indicating $\mathbf{z}$'s informative signal. Conversely, under VPL, the difference remains negligible during training. In this regime, Fig. 6b shows that the encoder's $\boldsymbol{\mu}$ and $\ell = \log \boldsymbol{\sigma}^2$ initially carry signal but soon drift toward $\boldsymbol{\mu} \approx 0$ and $\log \boldsymbol{\sigma}^2 \approx 0$, making the posterior almost the same as the prior. The encoded $\mathbf{z}$ lacks helpful information for the decoder's reward decision, resulting in a trivial solution that reduces the KL penalty to zero.

## B    JUSTIFICATION FOR P-IAF

For a prompt-response pair $(x, y_w, y_l)$ and a latent $\mathbf{z} \in \mathbb{R}^d$, let us define

$$\Delta r_\phi(\mathbf{z}) \triangleq r_\phi(x, y_w, \mathbf{z}) - r_\phi(x, y_l, \mathbf{z}),$$

which is the part inside the sigmoid $\sigma$ in Eq.(2). In the swap-guided base regularization, we regularize our encoder $q_\psi$ so that $\boldsymbol{\mu} = -\boldsymbol{\mu}_{\text{swap}}$ and $\boldsymbol{\sigma} = \boldsymbol{\sigma}_{\text{swap}}$. Assuming opposite coupling for the fictitious annotator $h_{\text{swap}}$, i.e., $\boldsymbol{\epsilon}_{\text{swap}} = -\boldsymbol{\epsilon}$, the latent samples become

$$\mathbf{z} = \boldsymbol{\mu} + \boldsymbol{\sigma} \odot \boldsymbol{\epsilon}, \quad \mathbf{z}_{\text{swap}} = \boldsymbol{\mu}_{\text{swap}} + \boldsymbol{\sigma}_{\text{swap}} \odot \boldsymbol{\epsilon}_{\text{swap}},$$

respectively. Consequently, we obtain

$$\mathbf{z}_0 = -\mathbf{z}_{0,\text{swap}}. \tag{14}$$

from the encoder $q_\psi$ in Fig. 4 under swapping. Finally, we regularize the base posterior for probability consistency

$$p_\phi(y_w \succ y_l \mid x, \mathbf{z}_0) = p_\phi(y_l \succ y_w \mid x, \mathbf{z}_{0,\text{swap}}). \tag{15}$$

However, unlike the base posterior $\mathbf{z}_0$, we cannot directly regularize after transforming $\mathbf{z} = \mathbf{z}_0$ into $\mathbf{z}_K$:

$$p_\phi(y_w \succ y_l \mid x, \mathbf{z}_K) = p_\phi(y_l \succ y_w \mid x, \mathbf{z}_{K,\text{swap}}), \tag{16}$$

because IAF entangles dimensions and contexts, so the posterior's mirrored structure need not be preserved after the flow. However, by supplying $\mathbf{c}_s$ and $\mathbf{c}_d$ to the $\mu_k$ and $\sigma_k$ functions as input

arguments separately, we can obtain similar results as base regularization indirectly. We will show it in this appendix. First, let us define swap probability error of transformed distribution $z_K$ by

$$\delta_p \triangleq \sigma\big(\Delta r_\phi(z_K)\big) - \sigma\big(-\Delta r_\phi(z_{K,\text{swap}})\big). \tag{17}$$

Further, we assume that (A1) $z \mapsto \Delta r_\phi(z)$ is Lipschitz; (A2) opposite coupling is used for the base noise, i.e., $\epsilon = -\epsilon_{\text{swap}}$; (A3) each step $k$-th scale function $\sigma_k$ is bounded by $\|\sigma_k(\cdot)\|_\infty \le \rho_k$;

*The key idea behind our justification of P-IAF is to demonstrate that the mirroring of preference swaps is realized in the transformed posterior $z_K$ by showing that the swap probability error $\delta_p$ of our P-IAF is smaller than that of IAF.*

**Lemma 1.** *Let us suppose that $z_0$ and $z_{0,swap}$ are warped to $z_K$ and $z_{K,swap}$ respectively, by P-IAF. Then, the swap probability error $\delta_p$ given in Eq.(17) is bounded by*

$$|\delta_p| \le \tfrac{1}{4}\,\delta_{r,K} + \tfrac{1}{4}\,L_r\,\|\delta_{z,K}\|. \tag{18}$$

*where the reward violation $\delta_{r,K} \triangleq \big|\Delta r_\phi(z_K) + \Delta r_\phi(-z_K)\big|$ and the latent mismatch $\delta_{z,K} \triangleq z_{K,swap} + z_K$.*

*Proof.* For all $a, b \in \mathbb{R}$, the logistic satisfies $|\sigma(a) - \sigma(b)| \le \tfrac{1}{4}|a - b|$, the swap probability error $\delta_p$ is bounded by

$$
\begin{aligned}
|\delta_p| &= \big|\sigma(\Delta r_\phi(z_K)) - \sigma(-\Delta r_\phi(z_{K,\text{swap}}))\big| \\
&\le \tfrac{1}{4}\big|\Delta r_\phi(z_K) + \Delta r_\phi(z_{K,\text{swap}})\big|.
\end{aligned}
$$

Using the triangle inequality, we obtain

$$
\begin{aligned}
|\delta_p| &\le \tfrac{1}{4}\big|\Delta r_\phi(z_K) + \Delta r_\phi(z_{K,\text{swap}})\big| \\
&= \tfrac{1}{4}\big|\Delta r_\phi(z_K) + \Delta r_\phi(-z_K) + \Delta r_\phi(z_{K,\text{swap}}) - \Delta r_\phi(-z_K)\big| \\
&= \tfrac{1}{4}\underbrace{\big|\Delta r_\phi(z_K) + \Delta r_\phi(-z_K)\big|}_{=\,\delta_{r,K}} + \big|\Delta r_\phi(z_{K,\text{swap}}) - \Delta r_\phi(-z_K)\big|
\end{aligned}
\tag{19}
$$

Further, using the Lipschitz assumption (A1),

$$\big|\Delta r_\phi(z_{K,\text{swap}}) - \Delta r_\phi(-z_K)\big| \le L_r\|z_{K,\text{swap}} - (-z_K)\| = L_r\|\delta_{z,K}\|.$$

Then we obtain Eq.(18) by combining reward violation and latent mismatch. $\square$

**Lemma 2.** *Let us suppose that the base posterior is given by $q_\psi(z_0 \mid \mathbb{D}_h) = \mathcal{N}(\mu, \sigma^2)$ and $q_\psi(z_0 \mid \mathbb{D}_{h_{swap}}) = \mathcal{N}(\mu_{swap}, \sigma^2_{swap})$. When we sample the latent $z_0$ and $z_{0,swap}$ based on assumption (A2), the base mismatch defined by*

$$\delta_{z,0} \triangleq z_{0,swap} + z_0 = (\mu + \mu_{swap}) + (\sigma - \sigma_{swap}) \odot \epsilon.$$

*And also bounded by*

$$\mathbb{E}\|\delta_{z,0}\| \le \|\mu + \mu_{\text{swap}}\| + \tfrac{1}{2}\,exp\,(\ell_{\max}^{(\infty)}/2)\,\|\ell - \ell_{swap}\|, \tag{20}$$

*where $\sigma = exp(\ell/2)$.*

*Proof.*

$$\mathbb{E}\|\delta_{z,0}\| \le \|\mu + \mu_{\text{swap}}\| + \|\sigma - \sigma_{\text{swap}}\|, \tag{21}$$

since $\mathbb{E}\|A\epsilon\| \le \sqrt{\mathbb{E}\|A\epsilon\|^2} = \|A\|$ where $A \triangleq \text{diag}(\sigma - \sigma_{\text{swap}}) \in \mathbb{R}^{d \times d}$.

Moreover, $\sigma = \exp(\ell/2)$, by the mean value theorem for $g(t) = \exp(t/2)$, then, $|g(a) - g(b)| = \tfrac{1}{2}\exp(\xi/2)|a - b| \le \tfrac{1}{2}\exp(\max\{a, b\}/2)|a - b|$ for some $\xi$ between $a$ and $b$.

$$\|\sigma - \sigma_{\text{swap}}\| \le \tfrac{1}{2}\exp(\ell_{\max}^{(\infty)}/2)\,\|\ell - \ell_{\text{swap}}\|, \qquad \ell_{\max}^{(\infty)} \triangleq \max\{\|\ell\|_\infty, \|\ell_{\text{swap}}\|_\infty\}. \tag{22}$$

$\square$

Hence, base regularization Eq.(7) directly decreases the base mismatch $\|\delta_{z,0}\|$.

From now on, we will compute the swap probability errors of our P-IAF and IAF methods one by one. Before deriving the swap probability errors of the two normalizing flows P-IAF and IAF, let us consider context vector $c$, which is an additional output of encoder $q_\psi$. For further development, we decompose the context vector $c$ into a swap-reversal context $c_d$ and swap-invariant context $c_s$ as:

$$c_d = \tfrac{1}{2}(c - c_{\text{swap}}), \qquad c_s = \tfrac{1}{2}(c + c_{\text{swap}}), \qquad c = c_d + c_s,$$

which ensures $c_{d,\text{swap}} = -c_d$ and $c_{s,\text{swap}} = c_s$.

**Assumption (A4).** Let $\mu_k, \sigma_k$ denote the $k$-th step shift and scale function. There exist non-negative constants [2]
$$L_{\mu,k}^z, \quad L_{\mu,k}^{c_d}, \quad L_{\mu,k}^{c_s}, \quad L_{\sigma,k}^z, \quad L_{\sigma,k}^{c_d}, \quad L_{\sigma,k}^{c_s}$$
such that, for all $(z, c_d, c_s)$ and $(z', c_d', c_s')$ in the valid input space,

$$\|\mu_k(z, c_d, c_s) - \mu_k(z', c_d', c_s')\| \leq L_{\mu,k}^z \|z - z'\| + L_{\mu,k}^{c_d} \|c_d - c_d'\| + L_{\mu,k}^{c_s} \|c_s - c_s'\|,$$
$$\|\sigma_k(z, c_d, c_s) - \sigma_k(z', c_d', c_s')\| \leq L_{\sigma,k}^z \|z - z'\| + L_{\sigma,k}^{c_d} \|c_d - c_d'\| + L_{\sigma,k}^{c_s} \|c_s - c_s'\|.$$

**Lemma 3** (Transformed mismatch of P-IAF). *Let us consider a normalizing flow P-IAF given by*

$$z_k = \mu_k(z_{k-1}, c_d) + \sigma_k(z_{k-1}, c_s) \odot z_{k-1}$$

*If we define the transformed mismatch $\delta_{z,k}$ at the $k$-th step as:*

$$\delta_{z,k} \triangleq z_{k,swap} + z_k$$

*Then, the mismatch $\delta_{z,k}$ of P-IAF is bounded by*

$$\|\delta_{z,k}\| \leq \left(\rho_k + L_{\mu,k}^z + L_{\sigma,k}^z \|z_{k-1}\|\right) \|\delta_{z,k-1}\|$$
$$+ \underbrace{\|\delta_{\mu,k}(c_d)\|}_{\text{swap-reversal violation } (\mu)} + \underbrace{\|\delta_{\sigma,k}(c_s)\|_\infty \|z_{k-1}\|}_{\text{swap-invariant violation } (\sigma)} \tag{23}$$

*where we define the $\mu$ swap-reversal violation and $\sigma$ swap-invariant violations at step $k$ as:*

$$\delta_{\mu,k}(c) \triangleq \mu_k(z_{k-1}, c) + \mu_k(-z_{k-1}, c_{swap}), \qquad \delta_{\sigma,k}(c) \triangleq \sigma_k(z_{k-1}, c) - \sigma_k(-z_{k-1}, c_{swap}). \tag{24}$$

*Proof.* In the P-IAF, the outputs from the $k$-th step is given by

$$z_k = \mu_k(z_{k-1}, c_d) + \sigma_k(z_{k-1}, c_s) \odot z_{k-1},$$
$$z_{k,\text{swap}} = \mu_k(z_{k-1,\text{swap}}, c_{d,\text{swap}}) + \sigma_k(z_{k-1,\text{swap}}, c_{s,\text{swap}}) \odot z_{k-1,\text{swap}}.$$

Then, the transformed mismatch $\delta_{z,k}$ at the $k$-th step is given by

$$\delta_{z,k} = \left[\mu_k(z_{k-1}, c_d) + \mu_k(-z_{k-1}, c_{d,\text{swap}})\right]$$
$$+ \left[\sigma_k(z_{k-1}, c_s) \odot z_{k-1} + \sigma_k(-z_{k-1}, c_{s,\text{swap}}) \odot (-z_{k-1})\right]$$
$$+ \left[\mu_k(z_{k-1,\text{swap}}, c_{d,\text{swap}}) - \mu_k(-z_{k-1}, c_{d,\text{swap}})\right]$$
$$+ \left[\sigma_k(z_{k-1,\text{swap}}, c_{s,\text{swap}}) \odot z_{k-1,\text{swap}} - \sigma_k(-z_{k-1}, c_{s,\text{swap}}) \odot (-z_{k-1})\right].$$

The first bracket equals $\delta_{\mu,k}(c_d)$ by Eq.(24), hence contributes $\|\delta_{\mu,k}(c_d)\|$.

For the second bracket, by Eq.(24) and $\|a \odot b\| \leq \|a\|_\infty \|b\|$:

$$\|\delta_{\sigma,k}(c_s)\|_\infty \|z_{k-1}\|.$$

For the third bracket, by (A4) in $z$:

$$\|\mu_k(z_{k-1,\text{swap}}, c_{d,\text{swap}}) - \mu_k(-z_{k-1}, c_{d,\text{swap}})\| \leq L_{\mu,k}^z \|\delta_{z,k-1}\|.$$

---

[2] The shift and scale networks are compositions of affine maps and smooth activations; hence they are locally Lipschitz on the working domain considered here.

For the fourth bracket, insert-delete $\sigma_k(z_{k-1,\text{swap}}, c_{s,\text{swap}}) \odot z_{k-1}$:

$$\sigma_k(z_{k-1,\text{swap}}, c_{s,\text{swap}}) \odot (z_{k-1,\text{swap}} + z_{k-1})$$
$$+ \big(\sigma_k(z_{k-1,\text{swap}}, c_{s,\text{swap}}) - \sigma_k(-z_{k-1}, c_{s,\text{swap}})\big) \odot (-z_{k-1}).$$

Bound the first term using (A3):

$$\|\sigma_k(z_{k-1,\text{swap}}, c_{s,\text{swap}}) \odot (z_{k-1,\text{swap}} + z_{k-1})\| \leq \rho_k \|z_{k-1,\text{swap}} + z_{k-1}\| = \rho_k \|\delta_{z,k-1}\|.$$

Bound the second term using (A4) in $z$:

$$\big(\sigma_k(z_{k-1,\text{swap}}, c_{s,\text{swap}}) - \sigma_k(-z_{k-1}, c_{s,\text{swap}})\big) \odot (-z_{k-1}) \leq L_{\sigma,k}^z \|\delta_{z,k-1}\| \|z_{k-1}\|$$

Collecting the bounds yields Eq.(23). $\qquad\square$

**Lemma 4** (Transformed mismatch of IAF). *Let us consider a normalizing flow IAF given by*

$$z_k = \mu_k(z_{k-1}, c) + \sigma_k(z_{k-1}, c) \odot z_{k-1},$$

*where $c = c_d + c_s$. Then, the mismatch $\delta_{z,k}$ of IAF is bounded by*

$$\|\delta_{z,k}\| \leq (\rho_k + L_{\mu,k}^z + L_{\sigma,k}^z \|z_{k-1}\|) \|\delta_{z,k-1}\|$$
$$+ \underbrace{\|\delta_{\mu,k}(c_d)\|}_{\textit{swap-reversal violation }(\mu)} + \underbrace{2L_{\mu,k}^{c_s}\|c_s\|}_{\textit{leak }(c_s \to \mu)} + \underbrace{\|\delta_{\sigma,k}(c_s)\|_\infty \|z_{k-1}\|}_{\textit{swap-invariant violation }(\sigma)} + \underbrace{2L_{\sigma,k}^{c_d}\|c_d\| \|z_{k-1}\|}_{\textit{leak }(c_d \to \sigma)}.$$

$$(25)$$

*Proof.* The derivation mirrors the proof in Eq.(23) except for two aspects. First, we replace $\delta_{\mu,k}(c_d)$ and $\delta_{\sigma,k}(c_s)$ with $\delta_{\mu,k}(c)$ and $\delta_{\sigma,k}(c)$ respectively using definition Eq.(24). Decompose these terms via insert–delete step:

$$\delta_{\mu,k}(c) = \underbrace{\mu_k(z_{k-1}, c_d, 0) + \mu_k(-z_{k-1}, c_{d,\text{swap}}, 0)}_{\delta_{\mu,k}(c_d)}$$
$$+ \underbrace{\big[\mu_k(z_{k-1}, c_d, c_s) - \mu_k(z_{k-1}, c_d, 0)\big]}_{\Delta_1^{(s)}} + \underbrace{\big[\mu_k(-z_{k-1}, c_{d,\text{swap}}, c_{s,\text{swap}}) - \mu_k(-z_{k-1}, c_{d,\text{swap}}, 0)\big]}_{\Delta_2^{(s)}},$$
$$\delta_{\sigma,k}(c) = \underbrace{\sigma_k(z_{k-1}, 0, c_s) - \sigma_k(-z_{k-1}, 0, c_{s,\text{swap}})}_{\delta_{\sigma,k}(c_s)}$$
$$+ \underbrace{\big[\sigma_k(z_{k-1}, c_d, c_s) - \sigma_k(z_{k-1}, 0, c_s)\big]}_{\Delta_1^{(d)}} + \underbrace{\big[\sigma_k(-z_{k-1}, 0, c_{s,\text{swap}}) - \sigma_k(-z_{k-1}, c_{d,\text{swap}}, c_{s,\text{swap}})\big]}_{\Delta_2^{(d)}}.$$

For the $\Delta$ terms, by the (A4),

$$\|\Delta_1^{(s)}\| \leq L_{\mu,k}^{c_s}\|c_s\|, \quad \|\Delta_2^{(s)}\| \leq L_{\mu,k}^{c_s}\|c_s\|,$$
$$\|\Delta_1^{(d)}\| \leq L_{\sigma,k}^{c_d}\|c_d\|, \quad \|\Delta_2^{(d)}\| \leq L_{\sigma,k}^{c_d}\|c_d\|.$$

Therefore, we obtain

$$\|\delta_{\mu,k}(c)\| \leq \|\delta_{\mu,k}(c_d)\| + 2 L_{\mu,k}^{c_s} \|c_s\|,$$
$$\|\delta_{\sigma,k}(c)\| \leq \|\delta_{\sigma,k}(c_s)\| + 2 L_{\sigma,k}^{c_d} \|c_d\|.$$

Collecting the bounds yields Eq.(25). $\qquad\square$

**Bound-Level Comparison between P-IAF and IAF**  Assume (A1)–(A4) hold and that P-IAF and IAF share the same architecture and training hyperparameters so that they admit the same upper bounds on the local Lipschitz constants $\{L_{\mu,k}^z, L_{\mu,k}^{c_d}, L_{\mu,k}^{c_s}, L_{\sigma,k}^z, L_{\sigma,k}^{c_d}, L_{\sigma,k}^{c_s}\}$, the same scale bounds $\rho_k$, the same reward Lipschitz constant $L_r$, and the same initial mismatch $\|\delta_{z,0}\|$. By Lemma 3 and Lemma 4, the IAF per-step bound contains two additional non-negative leak terms, $2L_{\mu,k}^{c_s} \|c_s\|$

and $2L_{\sigma,k}^{c_d} \|c_d\| \|z_{k-1}\|$, that are absent in P-IAF. Consequently, by induction over $K$ starting from Lemma 2,

$$\mathrm{UB}\big(\|\delta_{z,K}\|\big)^{(\text{P-IAF})} \leq \mathrm{UB}\big(\|\delta_{z,K}\|\big)^{(\text{IAF})},$$

where $\mathrm{UB}(\cdot)$ denotes the upper bound under the shared constants above.

Suppose in addition that the reward violation $\delta_{r,K}$ admits a common bound across the two flows, i.e., $\delta_{r,K}^{(\text{P-IAF})} \leq C_r$ and $\delta_{r,K}^{(\text{IAF})} \leq C_r$ for some $C_r \geq 0$. Combining this with Lemma 1, yields the following bound-level comparison.

$$\mathrm{UB}\big(|\delta_p|\big)^{(\text{P-IAF})} \leq \mathrm{UB}\big(|\delta_p|\big)^{(\text{IAF})}.$$

Thus, P-IAF attains a tighter bound on $|\delta_p|$ than the IAF bound.

**Remarks** (i) Swap-guided base regularization to encoder output reduces $\|\boldsymbol{\mu} + \boldsymbol{\mu}_{\text{swap}}\|$ and $\|\boldsymbol{\ell} - \boldsymbol{\ell}_{\text{swap}}\|$, thereby directly decreasing the expected base mismatch in Eq.(20). (ii) P-IAF's swap-guided split $(c_d \rightarrow \mu_k,\ c_s \rightarrow \sigma_k)$ eliminates leak terms in a shared context, tightening the swap-probability error bound.

## C    DETAILS OF ADAPTIVE LATENT CONDITIONING

We detail the adaptive latent conditioning applied in the decoder. As illustrated in Fig. 7a, the user-latent $z$ is mapped to a scale $\gamma$ and shift $\beta$ that modulate the incoming tokenized prompt-response embedding $e$ in a FiLM-style manner. Concretely, the decoder computes a latent conditioned representation by applying dimension-wise scaling and shifting to $e$ using $\gamma$ and $\beta$, respectively.

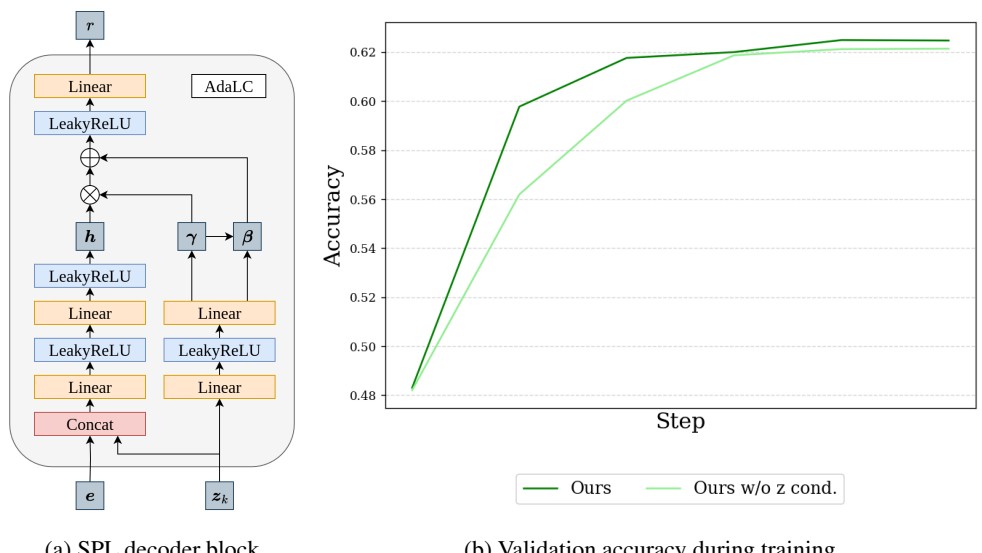

(a) SPL decoder block     (b) Validation accuracy during training

Figure 7: Details and training dynamics of the SPL decoder

Empirically, we also observe a training acceleration effect from adaptive latent conditioning. Fig. 7b reports preference-prediction accuracy evaluated periodically during training on *UF-P-4* with *Llama-3.1-8B*. The curve indicates that adaptive latent conditioning improves early-stage accuracy, suggesting that the reward model captures preferences more quickly with fewer samples. This is beneficial in data-scarce settings with minority preferences.

When the user-latent encodes preference information clearly (i.e., is low-uncertainty), the decoder leverages it via the modulation to personalize the reward. When the latent is uncertain or uninformative, the decoder naturally reduces the effective contribution of $z$, behaving closer to the base model. This adaptability ensures robustness across users with different feedback levels and consistency, while allowing strong personalization when reliable signals are available.

# D    ADDITIONAL EXPERIMENTS

**P-IAF vs. IAF**    We empirically test whether P-IAF yields better encoding performance than a standard IAF. To this end, we introduce two variants:

- **VPL-IAF**: An extension of VPL with a basic IAF posterior, used to examine the effect of a basic multi-modal posterior within a variational framework.

- **SPL-IAF**: Identical to SPL but replacing P-IAF with a standard IAF, serving as an ablation to evaluate the contribution of P-IAF.

These variants are not based on prior work; we design them specifically to compare our P-IAF with an IAF-based multi-modal posterior and to quantify the performance gains from P-IAF.

Table 4: Preference-prediction accuracy (%)

| Model | Method | UF-P-2 | UF-P-4 |
|---|---|---|---|
| | VPL | $62.37 \pm 0.15$ | $57.03 \pm 0.10$ |
| | VPL-IAF | $62.35 \pm 0.11$ | $58.01 \pm 0.43$ |
| Llama-3.2-3B | SPL-IAF | $63.09 \pm 0.14$ | $59.50 \pm 0.07$ |
| | SPL (Ours) | $\mathbf{63.28 \pm 0.13}$ | $\mathbf{61.56 \pm 0.03}$ |
| | VPL | $62.66 \pm 0.23$ | $57.14 \pm 0.05$ |
| | VPL-IAF | $62.82 \pm 0.13$ | $58.65 \pm 0.09$ |
| Llama-3.1-8B | SPL-IAF | $63.26 \pm 0.09$ | $60.56 \pm 0.34$ |
| | SPL (Ours) | $\mathbf{63.71 \pm 0.18}$ | $\mathbf{62.21 \pm 0.06}$ |

Simply adding IAF to VPL (VPL-IAF) does not yield robust encodings and often fails to prevent collapse. Likewise, replacing the P-IAF component in SPL with IAF (SPL-IAF) noticeably reduces accuracy. Nevertheless, SPL-IAF still achieves higher accuracy than VPL-IAF. Collectively, these results show that swap-guided base regularization and P-IAF effectively encode user preferences into identifiable user-latent variables.

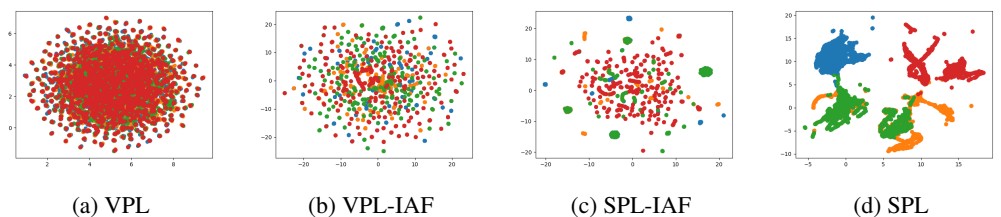

| (a) VPL | (b) VPL-IAF | (c) SPL-IAF | (d) SPL |

Figure 8: **Latent embeddings learned on the *UF-P-4* dataset.** We visualize 2D UMAP of latent embeddings $z$ from VPL, SPL, and their IAF variants (VPL-IAF and SPL-IAF), using Llama-3.1-8B to compare P-IAF against a standard IAF.

Fig. 8 visualizes the 2D UMAP projections of latent embeddings $z$ on the *UF-P-4* dataset. For VPL, the embeddings collapse into a single, non-identifiable cluster across preference types. For VPL-IAF, the embeddings do not collapse but remain scattered, concentrating in several dense regions. SPL-IAF likewise prevents collapse, with slightly lower accuracy than SPL. Its embeddings are less scattered than VPL-IAF while still constructing a complex posterior. In contrast, SPL most clearly separates preference types in latent space. Together, these observations suggest that a standard IAF allocates its modeling capacity to complex, unconstrained transformations of the latent space rather than to swap-derived structure. In contrast, our P-IAF preserves the expressivity of IAF while constraining it through swap-guided encoding, reducing unnecessary complexity and yielding more identifiable user-latent embeddings.

**Effect of Components**   We evaluate the effects of (i) base regularization $\mathcal{L}_{\text{guide}}$, (ii) P-IAF, and (iii) adaptive latent conditioning through an ablation on *UF-P-4* with *Llama-3.1-8B*. Table 5 summarizes results.

Table 5: Effect of each component

| $\mathcal{L}_{\text{guide}}$ | P-IAF | $z$ cond. | Acc. [%] | Active Units [%] |
|:---:|:---:|:---:|:---:|:---:|
| ✓ | | | 57.18 | 0.00 |
| | ✓ | | 59.10 | 11.03 |
| | | ✓ | 56.95 | 0.00 |
| ✓ | | ✓ | 56.87 | 0.00 |
| ✓ | ✓ | | 62.14 | 94.24 |
| | ✓ | ✓ | 62.08 | 93.07 |
| ✓ | ✓ | ✓ | **62.21** | **96.19** |

First, regarding the swap-guided base regularization ($\mathcal{L}_{\text{guide}}$), it was originally designed to guide P-IAF to learn an effective multi-modal posterior. Thus, the base regularization used alone without P-IAF, its effect naturally becomes modest because the model continues to generate a unimodal Gaussian posterior, which limits its ability to capture complex preference patterns. However, when combined with P-IAF, its effect is greatly enhanced: accuracy increases from 59.10% to 62.14%, and active units from 11.03% to 94.24%. We further evaluate swap-guided base regularization for preference encoding on the *UF-P-4* dataset using Llama-3.2-3B, comparing it with the baselines. The results are summarized in Table 6.

Table 6: Effect of the swap-guided base regularization

| Model | Method | Accuracy [%] | Active Units [%] |
|:---:|:---|:---:|:---:|
| | BTL | 57.07 | - |
| | VPL | 57.03 | 0.00 |
| Llama-3.2-3B | VPL w/ $\mathcal{L}_{guide}$ | 57.56 | 45.61 |
| | VPL-IAF | 58.01 | 39.45 |
| | SPL (Ours) | **61.56** | **82.32** |

The performance of VPL with base regularization lies between that of the original VPL and VPL-IAF. This shows that swap-guided base regularization improves VPL accuracy. Although improvement is smaller than that achieved by the IAF, this outcome is reasonable because the base regularization alone still produces a unimodal Gaussian posterior, while the IAF generates a multi-modal posterior. A unimodal posterior has limited capacity to capture complex preference structures, whereas a multi-modal posterior can represent richer and more diverse patterns. However, our design goal was not for swap-guided base regularization to serve as a strong preference encoder on its own, but rather to guide the construction of a multi-modal posterior in P-IAF by leveraging the structure of preference pair data. Nevertheless, the gains observed in some settings suggest that base regularization can also be effective as a stand-alone method for preference encoding.

Second, regarding adaptive latent conditioning, its primary role, as discussed in Appendix C, is to accelerate training and provide more stable optimization. Beyond these benefits, adaptive latent conditioning also helps models become more robust to noisy preference labels. To demonstrate this, we also evaluate robustness to noise, where the annotator occasionally selects the opposite or unrelated label. This type of noise can frequently arise in real-world settings. Concretely, on the *UF-P-4* dataset, we construct a noisy variant where 75% of preference pairs are unchanged and 25% are flipped, so the originally preferred response becomes less preferred. Table 7 shows results for Llama-3.2-3B.

Table 7: SPL accuracy and active units with noise on UF-P-4

| Model | Method | Accuracy [%] | Active Units [%] |
|---|---|---|---|
| | SPL | **61.41** | **82.62** |
| | SPL w/o z cond. | 58.08 | 80.37 |
| Llama-3.2-3B | SPL w/o $\mathcal{L}_{\text{guide}}$, z cond. | 56.92 | 0.00 |
| | VPL | 56.98 | 0.00 |

The results show that adaptive latent conditioning is essential for training noise-robust models. In the noisy setting, SPL achieves nearly the same preference-prediction accuracy as in the noise-free setting. Without adaptive latent conditioning, posterior collapse does not occur, but accuracy drops substantially. Moreover, if we additionally remove base regularization, the model exhibits collapse. For VPL, collapse occurs regardless of label noise. These findings show that swap-guided base regularization and adaptive latent conditioning together enable robust learning under noisy user feedback, making them essential for real-world settings with inherently noisy user choices. Across all ablations, these experiments show that combining these modules prevents collapse, produces informative user-latent variables, and achieves the best overall performance.

**Effect of P-IAF Depth**   We explore the effect of P-IAF depth $K$ on SPL. Each step updates the entire latent vector, allowing P-IAF to model high-dimensional structures using fewer steps. Using this property, we limit the range to shallow stacks and evaluate $K \in \{1, 2, 4\}$. Table 8 indicates that $K = 2$ yields the best performance. Thus, $K = 2$ is our default for all experiments. A single step prevents collapse and improves accuracy. With $K = 4$, performance drops, suggesting unnecessary expressivity reduces preference-prediction accuracy, similar to standard IAF.

Table 8: Effect of P-IAF depth $K$ on UF-P-4

| $K$ | Accuracy [%] | Active Units [%] |
|---|---|---|
| 1 | 60.58 | 91.60 |
| 2 | **62.21** | **96.19** |
| 4 | 61.92 | 93.16 |

**Preference Learning with fewer preference pairs**   We evaluate preference learning on the *UF-P-4* dataset when only a few preference pairs are provided to the model. Specifically, compared to the default setting, we randomly supply fewer pairs ($n \in \{2, 3, 4\}$) to *Llama-3.2-3B* and measure preference-prediction accuracy. As summarized in Table 9, SPL effectively encodes user preferences even under such limited preference signal. By contrast, VPL mitigates collapse with fewer pairs but captures user preferences poorly, resulting in accuracy similar to standard RLHF.

Table 9: Accuracy and active units with fewer preference pairs

| Model | Method | Accuracy [%] | Active Units [%] |
|---|---|---|---|
| | BTL | 56.94 | - |
| Llama-3.2-3B | VPL | 56.92 | 31.35 |
| | SPL (Ours) | **58.12** | **61.13** |

**Hyperparameter ablations**   We study the effect of the guidance loss hyperparameters $\lambda$ and $\eta$. The coefficient $\lambda$ controls the overall strength of the swap-guided base regularization term in Eq. (7), while $\eta$ balances the mean and standard deviation components within this loss. Their effects on preference-prediction accuracy on *UF-P-4* using Llama-3.2-3B are summarized in Tables 10 and 11.

Table 10: Effect of guidance loss weight $\lambda$

| $\lambda$ | Accuracy [%] | Active Units [%] |
|---|---|---|
| $1.0 \times 10^{-3}$ | $61.19 \pm 0.40$ | $81.05 \pm 4.98$ |
| $1.0 \times 10^{-4}$ | $60.90 \pm 0.20$ | $79.33 \pm 0.69$ |
| $1.0 \times 10^{-5}$ | $\mathbf{61.56 \pm 0.03}$ | $\mathbf{82.32 \pm 3.18}$ |
| $1.0 \times 10^{-6}$ | $59.74 \pm 0.09$ | $72.01 \pm 1.62$ |
| w/o $\mathcal{L}_{guide}$ | $58.53 \pm 0.13$ | $78.58 \pm 1.59$ |

Table 11: Effect of guidance balancing weight $\eta$

| $\eta$ | Accuracy [%] | Active Units [%] |
|---|---|---|
| 0.10 | $\mathbf{61.56 \pm 0.03}$ | $82.32 \pm 3.18$ |
| 0.25 | $61.49 \pm 0.11$ | $83.11 \pm 2.50$ |
| 0.50 | $61.18 \pm 0.18$ | $78.22 \pm 0.98$ |
| 1.00 | $61.44 \pm 0.06$ | $\mathbf{83.50 \pm 2.98}$ |

These results show that $\lambda = 1.0 \times 10^{-5}$ and $\eta = 0.1$ achieve the best performance. While $\eta$ affects accuracy only minimally, $\lambda$ has a more notable influence, which is expected since it directly controls the contribution of the guidance loss during optimization.

Table 1 shows that SPL is robust across a wide range of $\beta$ values, and Tables 10 and 11 indicate that coarse manual tuning of $\lambda$ and $\eta$ is sufficient to obtain stable performance. Thus, the overall tuning effort for SPL is no greater than that of standard preference learning methods.

# E   IMPLEMENTATION DETAILS

## E.1   HYPERPARAMETER SETTINGS

We detail the hyperparameters used in our experiments. Table 12 specifies the settings for generating the *Pets* and *UF-P* datasets. Table 13 specifies the training and evaluation hyperparameters. All experiments were run on a single NVIDIA RTX 4090 GPU and completed within two days.

Table 12: Hyperparameters for data generation

| Hyperparameter | Value |
|---|---|
| Token embedding dimension | 3072 (*Llama-3.2-3B-instruct*), 4096 (*Llama-3.1-8B-instruct*) |
| Max length | 1024 |
| Max preference pairs per sample $n$ | 8 |
| survey size for *UF-P* | 16 |
| Token data type | bfloat16 |
| Training samples in dataset | 4,000 (for *Pets*), 55,636 (for *UF-P-2*), 111,272 (for *UF-P-4*) |
| Evaluation samples in dataset | 400 (for *Pets*), 6,042 (for *UF-P-2*), 12,084 (for *UF-P-4*) |

Table 13: Hyperparameters for experiments

| Hyperparameter | Value |
|---|---|
| Encoder input dimension | 3072 (*Llama-3.2-3B-instruct*), 4096 (*Llama-3.1-8B-instruct*) |
| Latent dimension $d$ | 1024 |
| Learning rate | $1.0 \times 10^{-4}$ |
| Learning rate scheduler | cosine with 3% warm-up steps |
| Epoch | 2 |
| P-IAF flow step $K$ | 2 |
| Batch size | 32 (for *Pets*), 64 (for *UF-P*) |
| Optimizer | AdamW(with weight decay = 0.001) |
| KL Divergence weight $\beta$ | $1.0 \times 10^{-4}$ (for *Pets*), $3.0 \times 10^{-6}$ (for *UF-P*) |
| KL annealing scheduler | cosine cyclical from 0 to 1 (period = 10,000 steps) |
| Guidance loss weight $\lambda$ | $1.0 \times 10^{-5}$ |
| Guidance balancing weight $\eta$ | 0.1 |
| Active units threshold $\delta$ | 0.005 |

## E.2 ALGORITHMS

---
**Algorithm 1** Swap-guided Preference Learning (SPL)

---
**Require:** Preference Data $\mathbb{D} = \{\mathbb{D}_{h_1}, \cdots, \mathbb{D}_{h_m}\}$
**Require:** Encoder $q_\psi$, $K$-step P-IAF $F_{K_\psi}$, Reward Model $r_\phi$, prior $p(\boldsymbol{z}_k)$
 1: **while** not done **do**
 2:    Sample $\mathbb{D}_h = \{x^i, y_w^i, y_l^i\}_{i=1}^n \sim \mathbb{D}$
 3:    Tokenize $\boldsymbol{e}_{(\cdot)}^i = \text{LLM}^{\text{SFT}}(x^i, y_{(\cdot)}^i)$
 4:    Compute $\boldsymbol{\mu}, \boldsymbol{\ell}, \boldsymbol{c} = q_\psi(\{\boldsymbol{e}_w^i, \boldsymbol{e}_l^i\}_{i=1}^n), \boldsymbol{\mu}_{\text{swap}}, \boldsymbol{\ell}_{\text{swap}}, \boldsymbol{c}_{\text{swap}} = q_\psi(\{\boldsymbol{e}_l^i, \boldsymbol{e}_w^i\}_{i=1}^n)$
 5:    Sample $\boldsymbol{z}_0 \sim \mathcal{N}(\boldsymbol{\mu}, \boldsymbol{\ell})$
 6:    Compute $\boldsymbol{c}_d = \frac{1}{2}(\boldsymbol{c} - \boldsymbol{c}_{\text{swap}}), \boldsymbol{c}_s = \frac{1}{2}(\boldsymbol{c} + \boldsymbol{c}_{\text{swap}})$
 7:    Compute $\boldsymbol{z}_K = F_{K_\psi}(\boldsymbol{z}_0, \boldsymbol{c}_d, \boldsymbol{c}_s)$
 8:    Compute rewards: $r_w = r_\phi(\boldsymbol{e}_w, \boldsymbol{z}_K)$ and $r_l = r_\phi(\boldsymbol{e}_l, \boldsymbol{z}_K)$
 9:    Compute reconstruction loss: $\mathcal{L}_{\text{recon}} = -\log(\sigma(r_w - r_l))$
10:    Compute KL-loss: $\mathcal{L}_{\text{KL}} = \beta \cdot D_{\text{KL}}\big(\log q_\psi(\boldsymbol{z}_k \mid \mathbb{D}_h)\|p(\boldsymbol{z}_K)\big)$
11:    Compute guidance loss: $\mathcal{L}_{\text{guide}} = \lambda \cdot \left[\frac{1}{2}\big(1 + \cos(\boldsymbol{\mu}, \boldsymbol{\mu}_{\text{swap}})\big) + \eta \frac{1}{2}\big(1 - \cos(\boldsymbol{\ell}, \boldsymbol{\ell}_{\text{swap}})\big)\right]$
12:    Compute total loss: $\mathcal{L}_{\text{total}} = \mathcal{L}_{\text{recon}} + \mathcal{L}_{\text{KL}} + \mathcal{L}_{\text{guide}}$
13:    Update $E, q_\psi, F_{K_\psi}$ and $r_\phi$ by optimizing $\mathcal{L}_{\text{total}}$
14: **end while**

---

## F POTENTIAL AND SOCIAL EFFECTS

We address the tendency of standard RLHF to bias rewards toward majority preferences by encoding a user preference from a small number of comparisons and conditioning the policy on a user-latent, yielding a personalized policy $\pi_\theta(y \mid x, \boldsymbol{z}_k)$. We post-train the policy with

$$\max_{\pi_\theta} \mathbb{E}_{h \sim \mathbb{H}}\left[\mathbb{E}_{\substack{x \sim \mathbb{D} \\ y \sim \pi_\theta(y|x) \\ \boldsymbol{z}_k \sim q_\psi(\boldsymbol{z}_k|\mathbb{D}_h)}} \big[r_\phi(x, y, \boldsymbol{z}_k)\big] - \beta D_{\text{KL}}\big[\pi_\theta(y \mid x, \boldsymbol{z}_k) \,||\, \pi^{\text{SFT}}(y \mid x)\big]\right], \qquad (26)$$

which trains and deploys distinct behaviors conditioned on $\boldsymbol{z}_k$ inferred from the user's own choices. This differs from approaches that keep a single global policy and simply change the input $x$ via a user context: here, the conditional policy itself learns to act differently under different latents, rather than relying on prompt-only adaptation (Dong et al., 2022).

The scheme in Eq.(26) naturally extends to implicit-reward objectives such as Direct Preference Optimization (DPO) (Rafailov et al., 2023) by conditioning the policy and implicit reward surrogate on $\boldsymbol{z}_k$.

Plus, conditioning policy on user-latent is not limited to LLMs: Our swap-guided encoding and adaptive latent conditioning can be used when preferences are difficult to summarize, including in generative models or control settings (Poddar et al., 2024; Wang et al., 2025; Ng et al., 2025).

## G NOTATIONS

Table 14: Notations

| Notation | Meaning | Notation | Meaning |
|---|---|---|---|
| **Indices & counts** | | **Embeddings, latents & contexts** | |
| $i$ | preference-pair index | $\boldsymbol{e}_w$ | embedding of $y_w$ |
| $n$ | number of preference-pair in sample | $\boldsymbol{e}_l$ | embedding of $y_l$ |
| $N$ | total number of preference-pair | $\boldsymbol{z}$ | latent embedding |
| $j$ | dimension index | $\boldsymbol{\mu}$ | mean |
| $k$ | flow step | $\boldsymbol{\sigma}$ | standard deviation |
| $K$ | total flow step | $\boldsymbol{\ell}$ | log-variance |
| $d$ | latent dimension | $\boldsymbol{\epsilon}$ | random noise |
| | | $\boldsymbol{c}$ | shared context |
| | | $\boldsymbol{c}_d$ | swap-reversal context |
| | | $\boldsymbol{c}_s$ | swap-invariant context |
| **Users & sets** | | **Models & functions** | |
| $h$ | a user (annotator) | $q(\cdot)$ | encoder / variational posterior |
| $\mathbb{H}$ | user population | $r(\cdot)$ | decoder / reward function |
| $\mathbb{D}$ | full preference dataset | $p(\cdot)$ | preference probability |
| $\mathbb{D}_h$ | user $h$'s preference dataset | $f(\cdot)$ | autoregressive transform |
| $p$ | a preference type | $\mu_k(\cdot)$ | shift function at step k |
| $\mathbb{P}$ | set of preference types | $\sigma_k(\cdot)$ | scale function at step k |
| **Prompt & response** | | **Parameters & weights** | |
| $x$ | prompt | $\psi$ | learnable params (encoder & flow) |
| $y$ | response | $\phi$ | learnable params (decoder) |
| $y_w$ | chosen (winning) response | $\beta$ | KL-divergence weight |
| $y_l$ | rejected (losing) response | $\lambda$ | guidance loss weight |
| | | $\eta$ | guidance balancing weight |

**Norms** Throughout, $\| \cdot \|$ denotes the Euclidean norm for vectors and the Frobenius norm for matrices. We use $\| \cdot \|_\infty$ for the entrywise max norm when needed.

## THE USE OF LARGE LANGUAGE MODELS

We employed an LLM-assisted search to identify prior work on posterior collapse in VAEs and user-representation policies across domains. All retrieved items were manually reviewed by the authors to confirm their relevance before citation.

