# OpenReview forum: "Swap-guided Preference Learning for Personalized Reinforcement Learning from Human Feedback"
_ICLR.cc/2026/Conference — ICLR 2026 Poster_

### Official Review · Reviewer_cHBi · 2025-10-27

**Soundness:** 2
**Presentation:** 3
**Contribution:** 3
**Rating:** 6
**Confidence:** 2

**Summary:**

This paper introduces a new method for personalized RLHF. It includes an experimental evaluation on two Llama variants (3B, 8B), with the proposed method comparing favorably to the baselines.

**Strengths:**

The paper is well-written and easy to understand. It addresses an important and underserved niche of LLM personalization through diverse preferences. The method is well-justified and includes solid theoretical foundations.

**Weaknesses:**

1. In Section 3, the datasets are not yet introduced in any way, which makes it quite confusing to read.
2. No confidence intervals in Table 1.
3. Evaluations are based on completely synthetic preferences, not actual individual human preferences.
4. The paper does not provide the Pets dataset in meaningful detail, it also seems to be missing from the code release.
5. The performance improvement is relatively modest, which might be worth it depending on the computational cost trade-off.

**Questions:**

1. Can you run additional experiments to obtain tighter confidence bounds in Table 2? Some of the values are quite close to one another, making it hard to draw conclusions.
2. What is the computational overhead of using this method, over the baselines?
3. How exactly was the Pets dataset constructed?

---

> ### Author Response · Authors · 2025-11-21
>
> We sincerely thank you for the time and effort you put into reviewing our paper. Your constructive feedback has been incredibly helpful in refining our work. We address the weaknesses and your questions below.
>
> ### **Weaknesses**
>
> 1. **In Section 3, the datasets are not yet introduced in any way, which makes it quite confusing to read.**
>
>
>     We evaluate our method on two preference-learning benchmarks: Pets and UF-P.
>
>     Pets is a simple benchmark designed to test a method’s ability to distinguish between different preference distributions. It uses a single shared prompt, “Please talk about one kind of pets.” Each response is a diverse description of one of four animals: dog, cat, rabbit, or bird. Pets assumes two distinct user types with the following preference orders:
>
>     - Type 1: Bird > Dog > Cat > Rabbit
>     - Type 2: Bird > Cat > Dog > Rabbit
>
>     While the most preferred (Bird) and least preferred (Rabbit) pets are shared across both types, the middle rankings differ. For each type, the dataset contains 2,000 training and 200 test examples, for a total of 4,000 training and 400 test examples.
>
>     UF-P is a more complex benchmark that allows us to evaluate personalized reward model performance and scalability. It is derived from the UltraFeedback[1] prompt-response data, where responses are annotated by GPT-4 with scores for each user type. Specifically, UF-P-2 contains two preference types focusing on helpfulness and honesty, while UF-P-4 contains four preference types focusing on helpfulness, honesty, instruction-following, and truthfulness. The prompt-response pairs are grouped into sequences (up to the context length) and treated as the preference data of a single user.
>
>     For each preference type, UF-P provides 27,818 training and 3,021 test examples. Thus, UF-P-2 consists of 55,636 training and 6,042 test examples, and UF-P-4 consists of 111,272 training and 12,084 test examples. Due to its diverse preference modes and the wide variety of prompt-response pairs, the UF-P dataset is highly ambiguous and challenging.
>
>     To address the weakness, we have updated a brief mention of the dataset in Section 3: Motivation and a detail explanation of Pets in Section 5: Experiments.
>
>
> 2. **No confidence intervals in Table 1.**
>
>
>     We have revised our paper to report confidence intervals in Table 1. Thank you very much for this suggestion.
>
> 3. **Evaluations are based on completely synthetic preferences, not actual individual human preferences.**
>
>
>     We use synthetic preferences for two main reasons.
>
>     First, synthetic preferences are essential for isolating algorithmic behavior. By evaluating our model in a controlled setting where the ground-truth utility is known, we can clearly analyze the behavior and correctness of the proposed variational framework.
>
>     Second, we follow the experimental protocol used in our baseline VPL[2], which also relies on synthetic benchmark sets such as Pets and UF-P. Using the same synthetic benchmarks ensures fair and reproducible comparison with existing methods.
>
> 4. **The paper does not provide the Pets dataset in meaningful detail, it also seems to be missing from the code release.**
>
>
>     Regarding the details of the Pets dataset, we have now explained them more clearly in our response to your previous comment.
>
>     Regarding the code release, the generation command for Pets is already provided in our anonymous github. As written in the README:
>
>     `bash generate_llm_embeddings_pets.sh`
>
>     By running this command in a bash shell, you can generate the Pets dataset. We will update README with more detailed instructions to avoid any confusion.

---

> ### Author Response · Authors · 2025-11-21
>
> ### **Weaknesses**
>
> 5. **The performance improvement is relatively modest, which might be worth it depending on the computational cost trade-off.**
>
>
>     Before addressing it directly, we would like to clarify that our method should be compared with VPL, not VPL-IAF. VPL is the best-performing method among the previous works, and VPL-IAF corresponds to the previous VPL equipped with the standard IAF [3] proposed in this paper; thus, VPL-IAF does not exist in prior literature.
>
>     We included it in Table 2 only to illustrate the effect of incorporating standard IAF modules in the previous methods for ablation. We have removed the VPL-IAF and SPL-IAF rows from Table 2 and move them to the Appendix D: Additional experiments.
>
>     | Model | Method | Pets | UF-P-2 | UF-P-4 |
>     | --- | --- | --- | --- | --- |
>     | Llama-3.2-3B | BTL **[4]** | 57.48 ± 2.37 | 62.25 ± 0.03 | 57.07 ± 0.01 |
>     | Llama-3.2-3B | DPL **[5]** | 62.02 ± 1.92 | 62.22 ± 0.03 | 57.04 ± 0.05 |
>     | Llama-3.2-3B | VPL **[2]** | 99.67 ± 0.38 | 62.37 ± 0.15 | 57.03 ± 0.10 |
>     | Llama-3.2-3B | SPL (Ours) | **100.0 ± 0.00** | **63.28 ± 0.13** | **61.56 ± 0.03** |
>
>     | Model | Method | Pets | UF-P-2 | UF-P-4 |
>     | --- | --- | --- | --- | --- |
>     | Llama-3.1-8B | BTL **[4]** | 60.74 ± 0.49 | 62.59 ± 0.04 | 57.40 ± 0.28 |
>     | Llama-3.1-8B | DPL **[5]** | 61.03 ± 0.25 | 62.74 ± 0.03 | 57.66 ± 0.14 |
>     | Llama-3.1-8B | VPL **[2]** | 75.33 ± 0.63 | 62.66 ± 0.23 | 57.14 ± 0.05 |
>     | Llama-3.1-8B | SPL (Ours) | **100.0 ± 0.00** | **63.71 ± 0.18** | **62.21 ± 0.06** |
>
>     Without VPL-IAF and SPL-IAF in table, the performance improvements on Pets and UF-P-4 are clearly significant. For UF-P-2, the gains—from 62.37 ± 0.15 to 63.28 ± 0.13 with Llama-3.2-3B, and from 62.66 ± 0.23 to 63.71 ± 0.18 with Llama-3.1-8B—may appear “relatively modest,” but we believe these improvements are meaningful because SPL achieves them without requiring substantial additional computation or memory.
>
>     To support this point, we conducted additional experiments measuring computational and memory costs during training (for Llama-3.2-3B). The results are summarized in the table below.
>
>     | Method | sample/s | GPU hour | peak memory |
>     | --- | --- | --- | --- |
>     | VPL [2] | 6.070 | 13.363 | 6.25GB |
>     | SPL (Ours) | 5.952 | 13.590 | 6.65GB |
>
>     As shown in the table, our SPL achieves these gains with only minimal computational and memory overhead.
>
>
> ### **Questions**
>
> 1. **Can you run additional experiments to obtain tighter confidence bounds in Table 2? Some of the values are quite close to one another, making it hard to draw conclusions.**
>
>
>     As we explained in our response to Weakness 5, VPL-IAF and SPL-IAF are variations of our method and should be compared only in the ablation study. Therefore, after removing these two rows, we believe that the confidence intervals of VPL and our SPL do not overlap at all, even in UF-P-2. However, following your suggestion, we conducted experiments with two additional random seeds for UF-P to obtain tighter confidence bounds (table in weakness 5).
>
>     With these additional seeds, the standard deviation decreases in most cases. The updated results further confirm that SPL consistently achieves higher preference prediction accuracy than VPL.
>
>     For Pets, SPL outperforms VPL by 0.33% and 25.67% with Llama-3.2-3B and Llama-3.1-8B, respectively.
>
>     For UF-P-2, the gains are 0.91% and 1.05%, respectively.
>
>     For UF-P-4, the gains are 4.53% and 5.07% with Llama-3.2-3B and Llama-3.1-8B, respectively.
>
>     These results provide robust evidence that our proposed method is substantially more effective than baselines.
>
> 2. **What is the computational overhead of using this method, over the baselines?**
>
>
>     Please see our response to your previous comment (Weakness 5).
>
> 3. **How exactly was the Pets dataset constructed?**
>
>
>     Please see our response to your previous comment (Weakness 1).
>
>
> [1] Cui, Ganqu, et al. "Ultrafeedback: Boosting language models with high-quality feedback." (2023).
>
> [2] Poddar, Sriyash, et al. "Personalizing reinforcement learning from human feedback with variational preference learning." *Advances in Neural Information Processing Systems* 37 (2024): 52516-52544.
>
> [3] Kingma, Durk P., et al. "Improved variational inference with inverse autoregressive flow." *Advances in neural information processing systems* 29 (2016).
>
> [4] Ouyang, Long, et al. "Training language models to follow instructions with human feedback." *Advances in neural information processing systems* 35 (2022): 27730-27744.
>
> [5] Siththaranjan, Anand, Cassidy Laidlaw, and Dylan Hadfield-Menell. "Distributional preference learning: Understanding and accounting for hidden context in rlhf." *arXiv preprint arXiv:2312.08358* (2023).
>
>
> We hope this response clarifies your concerns and sufficiently answers your questions.

---

### Official Review · Reviewer_QmDu · 2025-10-30

**Soundness:** 3
**Presentation:** 4
**Contribution:** 3
**Rating:** 8
**Confidence:** 3

**Summary:**

This paper addresses the problem of posterior collapse in Variational Preference Learning for personalized reinforcement learning from human feedback, where user-specific latent variables often become uninformative and default back to a single reward model.
This paper propose Swap-guided Preference Learning (SPL), which leverages the mirroring property of swapped preferences and introduces three components: swap-guided base regularization, P-IAF, and adaptive latent conditioning.
Experiments show that SPL avoids collapse, stabilizes latent representations, and achieves higher preference-prediction accuracy than baselines.

**Strengths:**

1. The paper clearly identifies posterior collapse in personalized preference learning and introduces the intuitive idea of swap-guided mirroring, where swapping preferences flips the latent mean but keeps variance invariant, offering a novel and insightful diagnostic lens.
2. SPL integrates swap-guided base regularization, P-IAF, and adaptive latent conditioning into a coherent framework, directly addressing collapse while preserving user-specific information. The design is principled, interpretable, and builds effectively on established variational methods.

**Weaknesses:**

1. SPL introduces many additional hyper parameters, like $\beta, \gamma, \eta$, but does not analyze how robust these hyperparameters are or how much tuning would cost.

**Questions:**

1. How would you use the personalized reward model in policy model training?

---

> ### Author Response · Authors · 2025-11-21
>
> We are truly grateful for your encouraging evaluation and the time you dedicated to reviewing our work. We address your concerns regarding hyperparameters and clarify the usage of personalized reward model in policy training.
>
> ### **Weaknesses**
>
> 1. **SPL introduces many additional hyper parameters, like** $\beta$**,** $\gamma$ **and** $\eta$ **but does not analyze how robust these hyperparameters are or how much tuning would cost.**
>
>
>     Regarding the hyperparameters, we considered three key hyperparameters for SPL: $\beta$, $\lambda$, and $\eta$.
>
>     First, consider $\beta$, which controls the KL divergence term in Eq. (10). The change in accuracy for different $\beta$ values is already reported in Table 1 of the original paper. As discussed, while the baseline VPL [1] is highly sensitive to $\beta$, our SPL maintains a stable latent distribution across a wide range of $\beta$ values. We selected $\beta=3.0\times10^{-6}$ because it yielded the best accuracy in most cases, consistent with the setting used in VPL.
>
>     Second, $\lambda$ controls the strength of the guidance loss (base regularization) in Eq. (7).
>
>     Third, $\eta$ balances the mean and standard deviation terms within the guidance loss.
>
>     Their effects on accuracy are summarized in the tables below. (for UF-P-4, LLama-3.2-3B)
>
>     | Method | Accuracy [%] | Active Units [%] |
>     | --- | --- | --- |
>     | $\lambda=1.0\times10^{-3}$ | 61.19 $\pm$ 0.40 | 81.05 $\pm$ 4.98 |
>     | $\lambda=1.0\times10^{-4}$ | 60.90 $\pm$ 0.20 | 79.33 $\pm$ 0.69 |
>     | $\lambda=1.0\times10^{-5}$ | **61.56 $\pm$ 0.03** | **82.32 $\pm$ 3.18** |
>     | $\lambda=1.0\times10^{-6}$ | 59.74 $\pm$ 0.09 | 72.01 $\pm$ 1.62 |
>     | w/o $\mathcal{L}_{\text{guide}}$ | 58.53 $\pm$ 0.13 | 78.58 $\pm$ 1.59 |
>
>     | Method | Accuracy [%] | Active Units [%] |
>     | --- | --- | --- |
>     | $\beta=0.1$ | **61.56 $\pm$ 0.03** | 82.32 $\pm$ 3.18 |
>     | $\beta=0.25$ | 61.49 $\pm$ 0.11 | 83.11 $\pm$ 2.50 |
>     | $\beta=0.5$ | 61.18 $\pm$ 0.18 | 78.22 $\pm$ 0.98 |
>     | $\beta=1.0$ | 61.44 $\pm$ 0.06 | **83.50 $\pm$ 2.98** |
>
>     From these results, we observe that  $\lambda=1.0\times10^{-5}$ and $\eta=0.1$ achieve the best performance in our experiments. While $\eta$ affects accuracy only minimally, $\lambda$ has a more notable influence, which is expected since it directly controls the contribution of the guidance loss during optimization. We will include this table in the revised manuscript.
>
>     Regarding the tuning cost, we can say that the tuning cost of SPL is similar to that of the baseline VPL. Since VPL also has the hyperparameter $\beta$, SPL requires tuning only two additional parameters, $\lambda$ and $\eta$.  As shown in our Table 1 in paper and additional hyperparameter ablation, SPL is robust to a wide range of values for these parameters, and coarse manual tuning was sufficient to obtain stable performance. Therefore, the overall tuning effort for SPL does not exceed the typical tuning process used in existing preference learning methods. In response to this, we have revised our paper.
>
>
>
> ### **Questions**
>
> 1. **How would you use the personalized reward model in policy model training?**
>
>
>     Our personalized reward model can be directly plugged in as the reward function for post-training the policy.
>
>     The way we use SPL in policy training is identical to how VPL is used. Given a set of prompts, a human annotator selects the preferred response from each response pair. From all choices made by a given annotator, we infer that annotator’s user latent $z$, and we use this latent to train a personalized reward model that outputs rewards conditioned on $z$.
>
>     Due to this reward model produces different rewards for the same prompt-response pair depending on the latent, we can perform RLHF on the policy by feeding the same prompt-response together with different user latents $z$ and obtaining different reward values.
>
>     As a result, after post-training, the policy can produce different responses to the same prompt depending on the input latent. This procedure is exactly what Eq. (26) in Appendix F formalizes.
>
>     We hope this explanation clarifies our SPL policy training pipeline.
>
> **[1]** Poddar, Sriyash, et al. "Personalizing reinforcement learning from human feedback with variational preference learning." *Advances in Neural Information Processing Systems* 37 (2024): 52516-52544.
>
> Thank you again for your time and review. We hope our response has sufficiently resolved your concern and the question.

---

> > ### Comment · Reviewer_QmDu · 2025-11-25
> >
> > Dear authors, thank you for the response. I don't have further questions.

---

### Official Review · Reviewer_YXX7 · 2025-11-01

**Soundness:** 3
**Presentation:** 3
**Contribution:** 2
**Rating:** 4
**Confidence:** 4

**Summary:**

The paper tackles an important problem of personalizing RLHF to diverse users. It builds on existing paradigm that uses variational inference to model different users. The main focus of this paper is on the widely known “posterior collapse” problem with VAEs. The authors introduce swap annotators to the preference dataset i.e. effectively data augmentation by flipping the context labels and the target user preference order to create more data that voids posterior collapse. Further, they introduce a more expressive prior with a normalizing flow based model, and run experiments on preference modelling across two datasets. The experiments show that the final method is robust to posterior collapse and achieve a mean higher accuracy on preference modelling over the baselines.

**Strengths:**

- The paper is well written, with the motivation and background work sufficiently laid out.
- The authors clearly expose the posterior collapse problem with experimental evidence. This is a strong motivation towards explaining the issues with priors works and motivating the solutions introduced.
- The data augmentation technique for regularisation seems to be very effective and low-overhead, which potentially makes it very efficient.
- The P-IAF architecture introduced ensures that the regularisation and the mirror property hold after the transformations. Also present a theoretical justification for the approximation with the swap-invariant and swap-depedent variables.
- The authors show the validity of the method on multiple datasets againts multiple baselines.

**Weaknesses:**

- An issue with the method is that it assumes that the user-context is provided via binary preference labels. This doesnt seem to be scalable as more recent works [1], have focused on expanding user context to muti-turn dialogue. It would be interesting if the authors could discuss the applicability of the introduced regularisation to other forms of context.
- The swap based data augmentation seems to be a very interesting contribution. If the authors could include a baseline that trains a VPL based model with only the additional mirrored data, it could further show the benefit of the additional contributions beyond the swap guided pairs.
- In Table 3:, The adaptive latent conditioning and base regularisation provide negligible improvement to the modelling accuracy, which makes it hard to justify their contribution to the overall performance.
- Overall, the paper introduces interesting and know-techniques to improve preference modelling under diverse users. While the problem setting and the motivation of the solution is completely justified, the individual components introduced and the ablations over them provide relatively weak signals. If the authors are able to answer my questions, and provide additional ablations or experiments to strengthen their claims I would be happy to increase my score.

[1] Enhancing Personalized Multi-Turn Dialogue with Curiosity Reward. Yanming Wan, Jiaxing Wu, Marwa Abdulhai, Lior Shani, Natasha Jaques

**Questions:**

- In Figure 3b is the non-collapsed posterior trained via VPL or SPL? I.e does the sign reversal happen under the original dataset or the augmented dataset?
- The swap based regularisation assumes that the dataset contains context and preference pairs where the preference order is always dependent on the user context. But this would introduce wrong labels for preference pairs that are independent of the context i.e. same label regardless of the context. How do the authors resolve this issue?

---

> ### Author Response · Authors · 2025-11-21
>
> We sincerely thank you for dedicating your time to review our paper. Below, we present additional experiments addressing your concerns, and then respond to your questions.
>
> ### **Weaknesses**
>
> 1. **An issue with the method is that it assumes that the user-context is provided via binary preference labels. This doesnt seem to be scalable as more recent works (Wan et al., 2025), have focused on expanding user context to muti-turn dialogue.**
>
>
>     Our method intentionally uses preference-pair supervision. Preference pairs remain the most fundamental and widely used form of human feedback in preference-learning and RLHF pipelines because they are reliable, easy to annotate, and scalable.
>
>     Even in multi-turn dialogue settings, when a supervision signal is provided in the form of a preference pair between two responses, our SPL is applicable. Accumulating preference choices helps the model better characterize user preferences. In such scenarios, our SPL is particularly effective, as it encodes complex contextual preference information more accurately. As shown in our experiments, SPL learns user preferences more robustly under noisy labels (please see our reply to Weakness 3) and with limited preference contexts, which demonstrates strong potential in real-world user interaction scenarios.
>
>     For these reasons, we expect SPL to remain effective even in multi-turn dialogue settings, as long as preference pairs can be extracted from the dialogue history.
>
> 2. **The swap based data augmentation seems to be a very interesting contribution. If the authors could include a baseline that trains a VPL based model with only the additional mirrored data, it could further show the benefit of the additional contributions beyond the swap guided pairs.**
>
>
>     Following your suggestion, we conducted additional experiments in which we applied only the swap-guided base regularization (mirroring) to the previous method (VPL[1]**)**.
>
>     We performed these experiments on the UF-P-4 dataset using Llama-3.2-3B. We compared BTL, VPL, VPL-IAF, SPL, and VPL with mirroring. Please note that VPL-IAF corresponds to the VPL equipped with the IAF [2] proposed in this paper; thus, VPL-IAF does not exist in prior literature.
>
>     The results are summarized in the table below.
>
>     | Method | Accuracy [%] | Active Units [%] |
>     | --- | --- | --- |
>     | BTL **[3]** | 57.07 | - |
>     | VPL **[1]** | 57.03 | 0.00 |
>     | VPL w/ mirroring | 57.56 | 45.61 |
>     | VPL-IAF | 58.01 | 39.45 |
>     | SPL (Ours) | **61.56** | **82.32** |
>
>     From the results, we observe that the performance of VPL with mirroring lies between that of the original VPL and VPL-IAF. This indicates that the swap-guided base regularization (mirroring) indeed improves the accuracy of VPL. Although the improvement is smaller than that achieved by standard IAF, this outcome is reasonable because mirroring alone still produces a unimodal Gaussian posterior, while IAF generates a multi-modal posterior. A unimodal posterior has limited capacity to capture complex preference structures, whereas a multi-modal posterior can represent richer and more diverse patterns. We have included this in the revised manuscript.
>
> 3. **In Table 3, The adaptive latent conditioning and base regularisation provide negligible improvement to the modelling accuracy, which makes it hard to justify their contribution to the overall performance. The individual components introduced and the ablations over them provide relatively weak signals.**
>
>
>     First, regarding the swap-guided base regularization, this regularization was originally designed to guide P-IAF to learn an effective multi-modal posterior. Thus, when used alone without P-IAF, its effect naturally becomes modest because the model continues to generate a unimodal Gaussian posterior, which limits its ability to capture complex preference patterns. This explains why the improvement from the base regularization alone appears small. However, once combined with P-IAF, its effect becomes substantially stronger, as shown by comparing the results of $\mathcal L_{guide}$ + P-IAF with those of P-IAF alone in Table 3. In particular, the base regularization is essential for activating the latent variables (Active Units increasing from 11% to over 94%), which enables P-IAF to fully model multi-modal preference structures.
>
>     Second, the adaptive latent conditioning module was not intended to significantly increase accuracy by itself. Its primary role, as discussed in the paper, is to accelerate training and to provide more stable optimization. In addition, we show that this module contributes to robustness under noisy preference data (please see our reply to Question 2), which reflects more realistic scenarios.
>
>     Overall, these components play an enabling and stabilizing role rather than serving as standalone accuracy boosters. We have clarified this point in the revised manuscript.

---

> ### Author Response · Authors · 2025-11-21
>
> ## **Questions**
>
> 1. **In Figure 3b is the non-collapsed posterior trained via VPL or SPL? I.e does the sign reversal happen under the original dataset or the augmented dataset?**
>
>     Figure 3 is intended to motivate the swap-guidance mechanism introduced in this paper. In the previous method (VPL), we observed that applying a fictitious swap annotator reverses the sign of the latent posterior. We used this observation as a hint in designing our SPL framework. Therefore, the posterior shown in Fig. 3b is obtained from training the previous method (VPL), not SPL.
>
>     We have clarified this point in the revised manuscript.
>
>
> 1. **The swap based regularisation assumes that the dataset contains context and preference pairs where the preference order is always dependent on the user context. But this would introduce wrong labels for preference pairs that are independent of the context i.e. same label regardless of the context. How do the authors resolve this issue?**
>
>
>     We appreciate the reviewer for bringing up this point, as we did not explicitly consider robustness to noisy or inconsistent preference labels when submitting our SPL paper. However, following your suggestion, we conducted additional experiments by creating a noisy dataset in which 25% of the preference labels were randomly flipped and combined with the remaining 75% of the original UF-P-4 data. We then applied this noisy dataset to our baseline VPL and our SPL. We also tested the effect of our modules using Llama-3.2-3B. The results are summarized in the table below.
>
>     | Method | Accuracy [%] | Active Units [%] |
>     | --- | --- | --- |
>     | SPL (Ours) | **61.41** | **82.62** |
>     | SPL w/o z cond. | 58.08 | 80.37 |
>     | SPL w/o $\mathcal{L}_{\text{guide}}$, z cond. | 56.92 | 0.00 |
>     | VPL **[1]** | 56.98 | 0.00 |
>
>     These experiments simulate realistic user-interaction scenarios where preference annotations may contain mistakes, inconsistencies, or ambiguous signals. Interestingly, even though robustness was not explicitly built into the original design, SPL performs remarkably well under noisy supervision. Our modules, the adaptive latent conditioning and the swap-guided regularization contribute to stabilizing the latent representation and preventing collapse, resulting in more reliable performance compared to the baselines.
>
>     This outcome suggests that the structure of SPL naturally provides  robustness to noisy preference labels, even though this robustness was not an explicit design goal. We have included this observation and the new experimental results in the revised manuscript.
>
>
> **[1]** Poddar, Sriyash, et al. "Personalizing reinforcement learning from human feedback with variational preference learning." *Advances in Neural Information Processing Systems* 37 (2024): 52516-52544.
>
> **[2]** Kingma, Durk P., et al. "Improved variational inference with inverse autoregressive flow." *Advances in neural information processing systems* 29 (2016).
>
> **[3]** Ouyang, Long, et al. "Training language models to follow instructions with human feedback." *Advances in neural information processing systems* 35 (2022): 27730-27744.
>
> We thank you again for your valuable feedback and the great ideas that helped improve our paper. We hope these responses sufficiently address your concerns and questions.

---

### Official Review · Reviewer_yo5D · 2025-11-02

**Soundness:** 3
**Presentation:** 3
**Contribution:** 3
**Rating:** 6
**Confidence:** 4

**Summary:**

The paper shows that a leading personalization method, Variational Preference Learning, suffers from posterior collapse, causing user latents to be ignored. It proposes Swap-guided Preference Learning (SPL), which exploits a mirroring property of preference pairs by constructing fictitious “swap” annotators and enforcing sign-reversal of posterior means while keeping variances invariant. SPL combines (1) swap-guided base regularization, (2) a Preferential Inverse Autoregressive Flow (P-IAF) to disentangle swap-reversal and swap-invariant signals, and (3) adaptive latent conditioning for the reward decoder. Experiments demonstrate that SPL reliably prevents collapse across KL weights and improves preference-prediction accuracy over strong baselines.

**Strengths:**

1. The paper surfaces posterior collapse in preference learning (especially for VPL), provides detailed diagnostics of why the user latent is ignored, and introduces a swap-guided remedy centered on the mirrored “swap” property to keep the latent informative.
2. SPL is derived from a clear ELBO with a swap-guidance regularizer. Analyses show how base regularization and the proposed P-IAF reduce swap-mismatch and prevent cross-context leakage, which explains why the posterior should not collapse.
3. Across KL weights, SPL maintains substantially higher Active Units than VPL (particularly on UF-P-4), indicating non-collapsed, informative user latents.

**Weaknesses:**

1. Some recent methods reported results on UF-P and are discussed by the authors but not included in experiments (e.g., Nam et al., 2025), making it hard to situate SPL’s gains against the newest alternatives. Adding these would strengthen claims.
2. The approach adds flow-based inference (P-IAF) and swap-guidance terms, which plausibly increase compute and memory, but the paper does not report training time, inference latency, or budget-constrained comparisons. Reporting these costs would clarify practicality.

**Questions:**

1. What are SPL’s training and inference compute overheads relative to VPL/VPL-IAF?
2. Is SPL robust to noisy labels (e.g., label-flip noise) which is common in real-world settings?
3. How well does SPL generalize to unseen but in-distribution user contexts (i.e., train/test contexts are disjoint but user classes remain fixed)?
4. How does SPL compare to more recent personalized-reward baselines on UF-P (e.g., Nam et al., 2025)?
5. How were key hyperparameters chosen, and how did their values influence Active Units and accuracy?

---

> ### Author Response · Authors · 2025-11-21
>
> We sincerely thank you for your time and insightful review. Your feedback makes our paper more robust and complete. We address the weaknesses you pointed out and answer your questions.
>
> ### **Weaknesses**
>
> 1. **Some recent methods reported results on UF-P and are discussed by the authors but not included in experiments (e.g., Nam et al., 2025), making it hard to situate SPL’s gains against the newest alternatives. Adding these would strengthen claims.**
>
>
>     We agree that including results from the most recent methods, such as Nam et al. (2025)**[1]**, would further strengthen the comparison. However, although the paper is now available, its implementation has not been released yet, and the paper does not provide sufficient information to reproduce the method faithfully. Therefore, we are currently unable to include it in our experiments.
>
>     To ensure fairness and reproducibility, we restrict our comparisons to methods whose official code is publicly available or whose implementation details are sufficiently documented. Once the code for Nam et al. (2025) is released, our SPL can be evaluated under the same conditions.
>
>
> 1. **The approach adds flow-based inference (P-IAF) and swap-guidance terms, which plausibly increase compute and memory, but the paper does not report training time, inference latency, or budget-constrained comparisons. Reporting these costs would clarify practicality.**
>
>
>     Following your suggestion, we conducted additional experiments on UF-P-4 using Llama-3.2-3B. The experimental results are summarized in the table given below.
>
>     |  | train sample/s | train GPU hour | train peak memory | eval sample/s |
>     | --- | --- | --- | --- | --- |
>     | VPL **[2]** | 6.070 | 13.363 | 6.25GB | 4.044 |
>     | VPL-IAF | 6.040 | 13.411 | 6.32GB | 4.026 |
>     | SPL (Ours) | 5.952 | 13.590 | 6.65GB | 3.972 |
>
>     From the table, we can see that SPL introduces a modest amount of additional computational overhead compared to the baseline VPL. The GPU VRAM usage rises from 6.25 GB to 6.65 GB (a 6.4% increase) due to the P-IAF and adaptive latent conditioning modules. The total training time on the UF-P-4 dataset (~111k samples) increases from 13.363 hr to 13.590 hr, which corresponds to only about 13.6 minutes of additional computation. The evaluation speed decreases slightly from 4.044 samples/s to 3.972 samples/s, a reduction of approximately 1.8%, which is not substantial enough to affect practical applicability.
>
>     Overall, relative to the original VPL setup, this overhead represents only a very modest increase in computation and memory and is not significant in practice. Despite requiring only slightly more resources, SPL is able to capture meaningful latent distributions from subtle and even noisy preference signals, demonstrating the practical benefit of the additional modules. We have included this in the revised manuscript.
>
> ### **Questions**
>
> 1. **What are SPL’s training and inference compute overheads relative to VPL/VPL-IAF?**
>
>
>     Please see our response to your previous comment (Weakness 2).
>
> 2. **Is SPL robust to noisy labels (e.g., label-flip noise) which is common in real-world settings?**
>
>
>     We appreciate the reviewer for bringing up this point, as we did not explicitly consider robustness to noisy or inconsistent preference labels when submitting our SPL paper. However, following your suggestion, we conducted additional experiments by creating a noisy dataset in which 25% of the preference labels were randomly flipped and combined with the remaining 75% of the original UF-P-4 data. We then applied this noisy dataset to our baseline VPL and our SPL. We also tested the effect of our modules using Llama-3.2-3B. The results are summarized in the table below.
>
>     | Method | Accuracy [%] | Active Units [%] |
>     | --- | --- | --- |
>     | SPL (Ours) | **61.41** | **82.62** |
>     | SPL w/o z cond. | 58.08 | 80.37 |
>     | SPL w/o $\mathcal{L}_{\text{guide}}$, z cond. | 56.92 | 0.00 |
>     | VPL **[2]** | 56.98 | 0.00 |
>
>     These experiments simulate realistic user-interaction scenarios where preference annotations may contain mistakes, inconsistencies, or ambiguous signals. Interestingly, even though robustness was not explicitly built into the original design, SPL performs remarkably well under noisy supervision. Our modules, the adaptive latent conditioning and the swap-guided regularization contribute to stabilizing the latent representation and preventing collapse, resulting in more reliable performance compared to the baselines.
>
>     This outcome suggests that the structure of SPL naturally provides  robustness to noisy preference labels, even though this robustness was not an explicit design goal. We have included this observation and the new experimental results in the revised manuscript.

---

> ### Author Response · Authors · 2025-11-21
>
> ### **Questions**
>
> 3. **How well does SPL generalize to unseen but in-distribution user contexts (i.e., train/test contexts are disjoint but user classes remain fixed)?**
>
>
>     In datasets such as Pets, UF-P-2, and UF-P-4, the user classes (preference types) are fixed, but the contexts appearing in the test set are disjoint from those in the training set.
>
>     During evaluation, the model must infer the latent variable z from these unseen test contexts and correctly predict the user’s preference. Importantly, the user classes themselves are never provided to the model; they are only used for constructing the dataset split and for visualization.
>
>     Thus, training and test contexts are strictly separate, so the evaluation tests the model's ability to generalize to unseen but in-distribution contexts.
>
> 4. **How does SPL compare to more recent personalized-reward baselines on UF-P (e.g., Nam et al., 2025)?**
>
>
>     Please see our response to your previous comment (Weakness 1).
>
> 5. **How were key hyperparameters chosen, and how did their values influence Active Units and accuracy?**
>
>
>     Based on your suggestion, we conducted additional experiments to examine the effects of three key hyperparameters of SPL: $\beta$, $\lambda$, and $\eta$.
>
>     First, consider $\beta$, which controls the KL divergence term in Eq. (10). The change in accuracy for different $\beta$ values is already reported in Table 1 of the original paper. As discussed, while the baseline VPL is highly sensitive to $\beta$, our SPL maintains a stable latent distribution across a wide range of $\beta$ values. We selected $\beta = 3.0 \times 10^{-6}$ because it yielded the best accuracy in most cases, consistent with the setting used in VPL.
>
>     Second, $\lambda$ controls the strength of the guidance loss (base regularization) in Eq. (7).
>
>     Third, $\eta$ balances the mean and standard deviation terms within the guidance loss.
>
>     Their effects on accuracy are summarized in the tables below. (for UF-P-4, Llama-3.2-3B)
>
>     | Method | Accuracy [%] | Active Units [%] |
>     | --- | --- | --- |
>     | $\lambda=1.0\times10^{-3}$ | 61.19 $\pm$ 0.40 | 81.05 $\pm$ 4.98 |
>     | $\lambda=1.0\times10^{-4}$ | 60.90 $\pm$ 0.20 | 79.33 $\pm$ 0.69 |
>     | $\lambda=1.0\times10^{-5}$ | **61.56 $\pm$ 0.03** | **82.32 $\pm$ 3.18** |
>     | $\lambda=1.0\times10^{-6}$ | 59.74 $\pm$ 0.09 | 72.01 $\pm$ 1.62 |
>     | w/o $\mathcal{L}_{\text{guide}}$ | 58.53 $\pm$ 0.13 | 78.58 $\pm$ 1.59 |
>
>     | Method | Accuracy [%] | Active Units [%] |
>     | --- | --- | --- |
>     | $\beta=0.1$ | **61.56 $\pm$ 0.03** | 82.32 $\pm$ 3.18 |
>     | $\beta=0.25$ | 61.49 $\pm$ 0.11 | 83.11 $\pm$ 2.50 |
>     | $\beta=0.5$ | 61.18 $\pm$ 0.18 | 78.22 $\pm$ 0.98 |
>     | $\beta=1.0$ | 61.44 $\pm$ 0.06 | **83.50 $\pm$ 2.98** |
>
>     From these results, we observe that  $\lambda=1.0\times10^{-5}$ and $\eta=0.1$ achieve the best performance in our experiments. While $\eta$ affects accuracy only minimally, $\lambda$ has a more notable influence, which is expected since it directly controls the contribution of the guidance loss during optimization. We have included this in the revised manuscript.
>
> **[1]** Nam, Hyunji, et al. "Learning to summarize user information for personalized reinforcement learning from human feedback." *arXiv preprint arXiv:2507.13579* (2025).
>
> **[2]** Poddar, Sriyash, et al. "Personalizing reinforcement learning from human feedback with variational preference learning." *Advances in Neural Information Processing Systems* 37 (2024): 52516-52544.
>
> We hope these responses sufficiently address your questions. Thank you again for your valuable time and feedback.

---

### Author Response · Authors · 2025-12-03

We thank the reviewers for carefully reading our paper and for providing thoughtful feedback.

We identify posterior collapse in Variational Preference Learning (VPL) **[1]** and introduce Swap-guided Preference Learning (SPL) to learn expressive, robust, user-specific posteriors. SPL effectively reduces collapse and improves preference prediction.

Based on the reviews, the main strengths of our work can be summarized as follows:

1. The paper analyzes posterior collapse in prior methods and clearly explains its causes, then proposes solutions that directly address these issues.
2. The proposed method is theoretically well-justified and appears to be a potentially very effective technique.
3. Compared to prior methods, SPL is less sensitive to hyperparameters and encodes more informative user latents from preference pairs, thereby achieving higher performance.
4. The paper demonstrates the validity of the method on multiple datasets against multiple baselines.

Below is a summary of each reviewer’s main concerns and our responses.

1. The main concern of **cHBi (rating 6/confidence 2)** was that our performance improvements over VPL might appear modest.

    However, this concern arises from a misunderstanding of Table 2. Some of the compared methods (e.g., VPL-IAF and SPL-IAF) are not prior works but variations of our own method, included only for ablation. We moved these variants to the ablations in Appendix D to avoid confusion and clarified the correct comparison setting. Under the proper comparison (SPL vs prior methods), our method consistently achieves significant gains with minimal computational overhead.

2. The main concern of **QMDU (rating 8/confidence 3)** was that SPL introduces additional hyperparameters without sufficient analysis of their robustness or tuning cost.

    In response, we ran new ablations over $\beta$, $\lambda$ and $\eta$ and added them to Appendix D. These results show that our SPL is stable across a wide range of values, unlike prior work VPL, which is highly sensitive to $\beta$.

3. **YXX7 (rating 4/confidence 4)** is mainly concerned that swap-guided base regularization and adaptive latent conditioning yield only small accuracy improvements.

    Additional experiments show that both components offer clear benefits. Table 6 shows that swap-guided base regularization alone prevents collapse and improves preference-prediction accuracy. Table 7 further shows that the two modules keep SPL robust under realistic noise, allowing reliable encoding of user preferences even when some labels are flipped. In addition, adaptive latent conditioning also speeds up training. These results have been added to Appendix D.

    Reviewer YXX7 indicated they would raise their score if these concerns were addressed; we believe our new experiments and clarifications fully resolve them.

4. Reviewer **yo5D (rating 6/confidence 4)** raised an important concern about the computational and memory overhead of our SPL compared to the prior VPL.

    To address this, we additionally measured training time, inference speed, and GPU memory usage. The results show that SPL adds very small overhead. GPU memory increases by 0.4GB, total training time on UF-P-4 rises by only 13.6 minutes over a 13.3 hour run ($\approx$1.7% overhead), and inference slows by just 1.8%. For this minimal cost, SPL resolves posterior collapse seen in prior methods and learns identifiable, noise-robust latent distributions. These results show that our SPL is practical and scalable.


**[1]** Poddar, Sriyash, et al. "Personalizing reinforcement learning from human feedback with variational preference learning." *Advances in Neural Information Processing Systems* 37 (2024): 52516-52544.

We look forward to sharing SPL with the ICLR community.

Thank you.

---

### Meta-Review · Area_Chair_T1qw · 2025-12-29

**Summary:**

The reviewer's concerns are summarized as follows: the improvements based on Table 2 are marginal, the computational overhead relative to the baselines is significant, the robustness of the proposed method is questionable, the hyperparameter choice requires further consideration, and the components of the proposed method require clarification. Despite some misunderstanding, the authors provide clarification on the results and additional experiments in the rebuttal. After the rebuttal, these issues are mostly resolved. Based on the original reviews and the rebuttal, I recommend acceptance.

**Reviewer Concerns:**

The Reviewer yo5D's concerns regarding computational overheads and robustness to noisy labels or OOD scenarios are addressed. The concerns about baselines still need further clarification.

The Reviewer YXX7 concerns regarding scalability on multi-turn dialogue and the benefits of each component are addressed.

The Reviewer QmDu concerns regarding hyperparameters are addressed.

The Reviewer cHBi concerns that the details of the datasets are well-addressed.

**Reviewer Scores:**

The reviewer QmDu has participated in the discussion and will maintain the scores. The author provides sufficient details to address the reviewers' concerns.  However, the relative baselines cannot be answered without open source code. Hence, the yo5D will keep the scores. cHBi will also keep score due to the low confidence. Reveiwer YXX7 will raise scores due to the indication that he would raise scores if the concerns are addressed.

---

### Decision · Program_Chairs · 2026-01-26

Accept (Poster)